# The Role of Cullin-RING Ligases in Striated Muscle Development, Function, and Disease

**DOI:** 10.3390/ijms21217936

**Published:** 2020-10-26

**Authors:** Jordan Blondelle, Andrea Biju, Stephan Lange

**Affiliations:** 1Department of Medicine, University of California, La Jolla, CA 92093, USA; 2Department of Molecular and Clinical Medicine, University of Gothenburg, 41345 Gothenburg, Sweden

**Keywords:** cullin-RING ligase (CRL), Nedd8, protein degradation, ubiquitin-proteasome system (UPS), autophagy-lysosome system, striated muscle development, striated muscle function, muscular dystrophy, cardiomyopathy

## Abstract

The well-orchestrated turnover of proteins in cross-striated muscles is one of the fundamental processes required for muscle cell function and survival. Dysfunction of the intricate protein degradation machinery is often associated with development of cardiac and skeletal muscle myopathies. Most muscle proteins are degraded by the ubiquitin–proteasome system (UPS). The UPS involves a number of enzymes, including E3-ligases, which tightly control which protein substrates are marked for degradation by the proteasome. Recent data reveal that E3-ligases of the cullin family play more diverse and crucial roles in cross striated muscles than previously anticipated. This review highlights some of the findings on the multifaceted functions of cullin-RING E3-ligases, their substrate adapters, muscle protein substrates, and regulatory proteins, such as the Cop9 signalosome, for the development of cross striated muscles, and their roles in the etiology of myopathies.

## 1. Introduction

Cross-striated muscles, consisting of heart and skeletal muscles, account for 40 to 50% of the body mass in healthy individuals. Muscle mass is regulated by an equilibrium between protein synthesis and protein degradation. Similar to protein synthesis, protein degradation is a highly coordinated process that is required for cellular proteostasis as well as many cellular and molecular functions. The two main pathways that regulate protein degradation in cells, including striated muscle cells, are the autophagy-lysosome and the ubiquitin-proteasome system (UPS) (reviewed in [1,2]).

In striated muscle, the ubiquitin-proteasome pathway is responsible for the degradation of nearly 80% of proteins. The poly-ubiquitylation process is a three-step enzymatic cascade (reviewed in greater detail in [2,3,4,5]). The first step consists of the activation of the ubiquitin protein by the E1-activating enzyme. This step requires ATP to load the processed ubiquitin onto the E1-enzyme. The second step is catalyzed by the E2-conjugating enzyme and consists of the attachment of the activated ubiquitin to the E2 enzymes. Finally, E3-ligase enzymes catalyze the last step, attaching the ubiquitin protein to targeted substrate proteins. Once poly-ubiquitylated, substrates are typically sent to the 26S proteasome for degradation. However, E3-ligases may also promote other forms of substrate ubiquitylation (e.g., protein mono-ubiquitylation) with vastly different biological functions (reviewed in [4]).

Each step of the ubiquitylation cascade increases the specificity towards one or a subset of proteins. In mammals, only one E1 enzyme (Uba1) is responsible for the activation of the ubiquitin molecule. In comparison to the E1-enzyme, there are dozens of E2-enzymes and many hundreds of E3-ligases.

One of the largest groups of E3-ligases is formed by cullin proteins [6]. Cullins do not bind directly to their substrates but act as scaffold proteins that interact with a variety of accessory substrate adapter protein families (Figure 1A). In addition, the cullin E3-ligase complex needs to bind to one of two RING (really interesting new gene) finger-domain-containing proteins (Rbx1 or Rbx2) that have catalytic functionality for mediating transfer of ubiquitin onto substrates, and other regulatory proteins to become fully active (Figure 1B,C). While the cullin protein family consists of only eight members, namely cullin-1, cullin-2, cullin-3, cullin-4A, cullin-4B, cullin-5, cullin-7, and cullin-9/Parc [7], their modular assembly with accessory proteins into thousands of possible cullin E3-ligase complex combinations is the reason why they count among the largest family of E3-ligases in cells, thereby achieving broad substrate protein specificity [7].

Regulation of cullin activity is a complicated process involving actions of the Cop9 (constitutive photomorphogenesis 9) signalosome complex, accessory proteins like Cand1 (cullin-associated neddylation-dissociated; also called Tip120 (TBP-interacting protein 120)), and the ubiquitin-like modifier Nedd8 (neuronal precursor cell-expressed developmentally downregulated protein 8; Figure 1C). Only Nedd8-modified cullin complexes are catalytically active. The process of cullin neddylation is a three-step enzymatic cascade that bears a high resemblance to the ubiquitylation cascade. Cullins can be deneddylated by the action of the Cop9 signalosome complex.

While the most prominent function of cullin E3-ligases lies in regulating the cell cycle, mainly through the degradation of cyclins [8,9], several cullins and their muscle-specific substrate adapters and identified substrate proteins have recently been shown to be crucial for cross-striated muscle development and function. We will describe below in detail the role that cullin-RING ligases (CRLs) and their associated proteins play in muscle homeostasis, development, and disease. We place special emphasis on summarizing available data when mutations in CRLs, their substrate adapters, and regulatory proteins, as well as substrates, are associated with the etiology of skeletal muscle myopathies or cardiomyopathies.

## 2. Muscle-Specific Regulation of CRL Function

CRL function is regulated on multiple levels. While all cullin proteins display a ubiquitous expression pattern, protein levels of their substrate adapters can vary widely within different tissue- and cell-types. As the accessory substrate adapters determine which proteins are subjected to cullin E3-ligase dependent ubiquitylation, their availability in each given tissue or cell-type may become rate-determining.

Besides this innate restriction on the type of available CRL complexes, cullin E3-ligase activity is also determined by other accessory protein complexes, posttranslational cullin modifications, and actions of the Cop9 signalosome complex. Some of these regulatory mechanisms are found in all cell- and tissue-types, while others appear to be specific adaptations for cross-striated muscles.

### 2.1. Nedd8

Neddylation is the biochemical process by which the ubiquitin-like modifier Nedd8 (neuronal precursor cell-expressed developmentally downregulated protein 8) is conjugated to its target proteins. Nedd8 is an 81 amino acid polypeptide that is very similar to the ubiquitin protein, suggesting a close evolutionary relationship. Indeed, Nedd8 is the closest relative of ubiquitin among all ubiquitin-like proteins. The *Nedd8* gene is developmentally down-regulated and, in adulthood, becomes almost exclusively expressed in muscle tissues (i.e., heart and skeletal muscles) [14,15]. Neddylation is a posttranslational modification characteristic for all cullin proteins and stabilizes cullin-RING E3-ligases in a conformation that promotes substrate ubiquitylation, therefore modulating CRL activity and consequently substrate–protein degradation [16]. While the main targets of Nedd8 are the cullins, recent reports show that non-cullin proteins may also be neddylated, a process named “non-cullin neddylation” [17]. However, the authors of this manuscript stress that many of the reported Nedd8 target proteins identified in the literature use over-expression experiments, which may give rise to experimental artifacts and false-positive neddylation of substrate proteins. Altering the cellular balance between Nedd8 and ubiquitin allows for the utilization of the enzymatic ubiquitylation cascade due to the evolutionary closeness between these two modifiers. Hence, guidelines for the identification of genuine protein neddylation were proposed that circumvent the overexpression artifact [17,18].

Ablation of neddylation and subsequent CRL activity through Mln4924 (Pevonedistat), a Nedd8 activating enzyme (Nae1, also known as Appbp1) inhibitor [19], triggers muscle cell differentiation defects in vitro [20,21]. Indeed, inhibition of CRL activity in C2C12 blocks myoblast fusion and differentiation (Figure 2A), as shown by a decrease of both fusion index and expression of myogenic factors, such as sarcomeric myosin and myogenin [20]. Moreover, inhibition of Nedd8 impairs normal acetylcholine receptor (AChR) clustering upon neural agrin stimulation, suggesting an important role of CRL activity for neuromuscular junction development and establishment [20,21]. Data from siRNA of Nedd8 in vitro phenocopy the inhibitor results, indicating the high specificity of Mln4924 [20]. Intriguingly, the removal of Mln4924 releases the block on C2C12 differentiation, allowing for the delayed formation of myotubes [20]. Summarized, data on the inhibition of neddylation through Mln4924 and siRNA point towards crucial roles for CRL activity during skeletal muscle formation and neuromuscular junction development.

Intriguingly, short term exposure of Mln4924 has no discernable effects on myofilament structures in neonatal mouse cardiomyocytes when added 48 h after plating of cells in vitro (Figure 2B), while leading to a reversible cardiomyopathy in vivo [22]. The authors of this study show specific effects of Mln4924 on the postnatal maturation of rat hearts, including reduced cardiomyocyte proliferation during early postnatal development. While the effects of short term Mln4924 in neonatal animals resulted in a mild hypertrophy in adulthood, they also sensitized treated rats to isoproterenol, exacerbating the effects of this β-adrenoreceptor agonist [22]. However, short-term exposure of Mln4924 to older animals did not result in noticeably altered impairment of cardiac function [23], suggesting more crucial roles of CRLs during early cardiac development.

CRLs are involved in the degradation of cyclin proteins and play a significant role in cancer development. Thus therapies that utilize Mln4924 to target the function of these E3-ligases for cell-proliferation are very appealing. Hence, a series of phase I/II/III clinical trials using Mln4924 (Pevonedistat) is currently under evaluation for the treatment of various cancer forms [24,25,26,27,28]. So far, data regarding the side effects of Pevonedistat treatment on skeletal muscle or cardiac tissues in treated patients are sparse. One study [29] showed increased muscle cramps as an adverse event in treated patients, which led to the decision to no longer dose patients above 110 mg/m^2^. Pevonedistat treatment also resulted in sinus tachycardia in some individuals in this clinical trial [29]. Given the data obtained in vitro [20] and rodent models [22,23], close monitoring of cross-striated muscle tissues in patients treated with Pevonedistat is necessary and requires further attention. However, some of the drug effects may be reversible once the treatment is stopped.

### 2.2. Nae1/Appbp1 and Uba3/Nae2

The process of cullin neddylation involves a series of enzymatic reactions with great similarities to the ubiquitylation cascade. Two proteins that are critical in the activation and transfer of Nedd8 onto cullin are the Nedd8 activating enzymes Nae1/Appbp1 and Nae2/Uba3. Both proteins interact to form a functionally active holoenzyme. The activity of Nae1 can be efficiently and reversibly inhibited by Mln4924 [19,20].

Conditional knockout of *Nae1* in adult excitatory forebrain neurons led to synapse maturation, stability, and function defects, including impaired neurotransmission. These data suggest crucial roles for protein neddylation in synaptogenesis in vivo [31]. While these results demonstrate the importance of neddylated CRL for neuronal synapses, they may also suggest potential roles for Nae1 functions in the development and function of motorneurons and the neuromuscular junction. Indeed, in vitro data from Nae1 inhibition by Mln4924 in C2C12 support this hypothesis [20]. However, these results will need further in vivo experiments for confirmation.

Recently, the generation of cardiac specific *Nae1* knockouts uncovered the primordial role of neddylation for ventricular chamber maturation [15]. These mice develop myocardial hypoplasia, ventricular noncompaction, and heart failure leading to perinatal death, within 72 h. Molecularly, the ablation of Nae1 in cardiac cells leads to the accumulation of the hippo kinases Mst1 (also known as Stk4 (serine/threonine kinase 4)), Lats1/2 (large tumor suppressor kinase 1 and 2), suggesting an important developmental function of neddylation for the essential modulation of the hippo-YAP (Yes1-associated transcriptional regulator) signaling pathway [15].

Global knockout of *Uba3* (ubiquitin-like modifier activating enzyme 3, also known as Nae2), a catalytic subunit of the Nedd8 activating enzyme complex that also contains Nae1, leads to embryonic lethality at the pre-implantation stage in mice [32]. The cell cycle is impaired in these embryos with no entry into the S-phase, likely due to the aberrant expression and defective degradation of cyclin E [32]. The early lethality in this animal model does not allow investigations of a potential muscle phenotype. To date, there are no data available on a conditional cardiac or skeletal muscle-specific *Uba3*-knockout.

### 2.3. The Cop9 Signalosome

The Cop9 signalosome (constitutive photomorphogenesis 9) is a highly conserved protein complex constituted of 8 subunits named Csn1-8. First identified in *Arabidopsis thaliana* to be important for photomorphogenesis [33,34], this highly conserved 300+ kDa multi-protein complex was later found to be a master regulator of cullin E3-ligase de-activation. The Csn complex has structural similarity to the 26S proteasome lid and eukaryotic translation initiation factor (eIF3) complexes [35,36,37], and some of its subunits have evolutionarily conserved domains that are found in all three structures. Two core Csn subunits that form a globular heterodimer, Csn5 (also known as Jab1, Cops5) and Csn6 (also known as Cops6), contain both a Mpr1/Pad1 N-terminal (MPN) domain. However, only the MPN-domain in Csn5 contains a zinc ion coordinating the JAMM (Jab1/MPN/Mov34 metalloenzyme) motif, which allows the protein to possess isopeptidase activity. The other Csn subunits contain PCI (proteasome, Cop9 signalosome, translation initiation factor) domains, which can also be found in eIF3 proteins that bind to DNA/RNA [38]. Over the last years, important strides have been made to unravel the molecular basis for the Cop9 signalosome biology, first when the full crystal structure of the complex became available [39], and more recently, when the basis for the structural interaction between cullin and the Cop9 signalosome was revealed [30,40] (Figure 2C). These seminal manuscripts outline the critical steps by which the Cop9 signalosome deneddylates cullin proteins. In the first step, Csn subunits 2 and 4 clamp onto Rbx1 and the C-terminal region of the active Nedd8-modified cullin E3-ligase. Through a series of structural rearrangements, Csn5 and Csn6 move into close spatial proximity of the Nedd8 attachment site within cullin. Finally, the catalytic isopeptidase activity in the Csn5 subunit cleaves the covalent bond between Nedd8 and its attached lysine residue in cullin, thereby releasing Nedd8 and inactivating the E3-ligase.

Global loss of Csn subunits leads to early embryonic lethality [38], necessitating the use of tissue-specific knockouts to study their function in the heart and skeletal muscles. Cardiac specific knockouts of *Csn8* (also known as Cops8) led to a progressive dilated cardiomyopathy resulting in heart failure and death of all mice within 52 days [41]. On the molecular level, loss of Csn8 led to a destabilization of the Cop9 signalosome holocomplex and reduction in protein levels of most Csn subunits. Concomitant with the loss of Cop9 activity is an increase in E3-ligase activity, best seen in the increase of poly-ubiquitylated proteins alongside elevated levels of proteasome subunits and heat-shock proteins in cardiac extracts of *Csn8* knockout mice. However, this increased E3-ligase activity may have been caused by the upregulation of RING-type E3-ligases (such as mouse double minute 2 homolog (Mdm2) or the muscle ring-finger protein 1 (MuRF1)), while CRL adapter proteins were decreased (Fbxo32/Atrogin-1, VHL, or β-Trcp). Indeed, the authors found no accumulation of well-characterized CRL substrates, such as Hif1-α, calcineurin, or β-catenin. Ultimately, *Csn8* knockout cardiomyocytes showed increased membrane damage, resulting in significant leukocyte infiltration and necrosis of the cells. Follow-up studies indicated that cardiomyocyte necrosis in *Csn8* knockouts is accompanied by impaired autophagic flux [42,43], toxic accumulation of protein aggregates (proteotoxicity) [44], and that key factors in the necroptotic pathway play a role in the death of cardiomyocytes, as inhibition of receptor interacting serine/threonine kinase 1(Ripk1) by necrotstatin-1 or haploinsufficiency of receptor interacting serine/threonine kinase 3 (Ripk3) prolonged median life-span of knockout animals [45].

While Csn subunits are more stable in the Cop9 signalosome holocomplex [41,46], they are found in up to 35 smaller sub-complexes [35] and even exhibit independent cellular functions [47,48]. Indeed, multiple subcellular localizations for varying Csn subunits have been described in mammalian and invertebrate skeletal muscle cells and cardiomyocytes, including the nucleus [41], the sarcolemma and t-tubules [47,49], the mitochondria [49], the sarcomere [49,50], or the intercalated disk [51]. We find that Csn subunits also exhibit differing subcellular localizations when investigated in cultured neonatal mouse cardiomyocytes and adult cardiac tissues (Figure 2D,E). The spatial and temporal localization is perhaps reflective of the varying and diverse canonical and non-canonical functions of individual Csn subunits, the complete signalosome, or smaller Csn sub-complexes in cross striated muscle cells [35]. In any case, the spatial distribution is influenced by interactions of the holocomplex to cullin E3-ligases and other binding partners that associate with individual Csn subunits. The catalytically active Csn5/Jab1 subunit was found to interact independently of other Csn proteins with cardiac L-type calcium channels [47], localizing both proteins to sarcolemmal membranes and transverse T-tubules in the heart and regulating the activity of the channels. Another function for Csn5 that seems independent of its role in the Cop9 signalosome is its interaction with endothelin type A and B receptors [48], as well as the heart and neural crest derivatives expressed Hand2 transcription factor [52]. These interactions suggest a role for Csn5 in the regulation of cardiac signaling pathways and the cardiac gene program.

Another report found binding of Csn3 (also known as Cops3) to muscle β1D-integrin, tethering this Cop9 subunit to costameres overlying the sarcomeric Z-disk [49], while Csn6 was reported to maintain desmosome structure at the cardiomyocyte intercalated disk [51].

While more information is available on the role of Csn subunits and the Cop9 signalosome in the heart, several reports also suggest important roles for the development and function of skeletal muscles. Loss of Csn3 in C2C12 was found to impair myoblast proliferation and differentiation of the cells into myotubes. The authors of this manuscript specifically suggest that Csn3 knockdown alters cellular NF-κB signaling, leading to its nuclear accumulation [46]. Hunter and co-authors investigated changes to several Cop9 subunits in undifferentiated and differentiated C2C12. Csn3 was evenly distributed between the cytoplasm and the nucleus in myoblasts and becomes transiently upregulated during differentiation into myotubes, which coincided with the redistribution of the protein from the nucleus to the membrane [49]. The same authors saw changes to the subcellular localization to nearly all Csn subunits during C2C12 differentiation, some also exhibiting changes in isoform expression.

Another study investigated a role for Csn2 (also known as Cops2) in a rodent cancer cachexia model [53]. Csn2 was downregulated in muscles of Lewis lung carcinoma bearing rodents, a result that was later confirmed also for skeletal muscles in cancer patients. On the molecular level, loss of Csn2 in human myotubes led to the deregulation of 872 genes, influencing pathways that modulate the actin cytoskeleton and cellular metabolism and catabolism. Intriguingly, Csn2 was among only nine proteins whose expression was normalized by exercise, a known non-pharmacological treatment to counter the cachexic effects of cancer therapy.

Pharmacological intervention of Cop9 signalosome activity is a relatively new therapeutic target to fight cancer. Csn5i-3 is a newly developed inhibitor of Csn5 deneddylation activity (Table 1). It was shown to keep CRL E3-ligases in the neddylated (active) state. Counterintuitively, chronic neddylation of CRLs leads to the inactivation of a subset of these E3-ligases by inducing degradation of their adapter proteins [54]. Another small compound that was found to affect Csn5 activity is doxycycline [55]. Although high concentrations of the drug were required to block the isopeptidase function of Csn5 (IC_50_: ~110 µM), tested lymphoma cells enriched doxycycline up to 40-fold compared to the concentration added to the culture media.

### 2.4. Senp8/Nedp1 and Other Nedd8-Modifying Enzymes

Besides actions of the Cop9 signalosome, cullin E3-ligases can also be deneddylated by Senp8/Nedp1 (also called Den1 human deneddylase 1), a sumo (small ubiquitin-like modifier) peptidase family member with specificity to Nedd8 [72,73]. While this enzyme displays only moderate mRNA levels in the heart and low levels in skeletal muscles, it was shown to act on neddylated muscle AChR subunits in vitro, potentially reversing the neddylation of these important neuromuscular junction proteins caused by the E3-ligase Rapsyn [21]. Besides acting on Nedd8, Senp8 is also able to de-sumoylate substrate proteins, such as Fog2 (friend of gata 2), a transcription factor important for heart morphogenesis [74]. Recently, forced Nedp1 expression was demonstrated to counteract proteotoxic stress in cardiomyocytes [75]. This approach may possibly serve as a novel therapeutic strategy that overcomes problems with modifying Cop9 signalosome activity.

Besides Nedp1, Uchl3 (ubiquitin C-terminal hydrolase L3) and Usp21 (ubiquitin specific peptidase 21) are two more enzymes that have been demonstrated to act on Nedd8 [76,77]. Usp21 is more similar to Senp8 in its functionality by deneddylating Nedd8-modified substrate proteins. Uchl3, on the other hand, appears to be more important for processing the C-terminus of newly synthesized Nedd8, a reaction that is important for subsequent addition of Nedd8 to substrate proteins through actions of Nae1 and Uba3. Intriguingly, *Uchl3* knockout mice display metabolic changes to their skeletal muscles, which were characterized by increased AMP-activated protein kinase (Ampk) activity and β-oxidation [78]. Further analysis revealed the accumulation of poly-ubiquitylated proteins and subsequent activation of the unfolded protein response [79]. However, it is unknown whether this increase in poly-ubiquitylation was caused by the augmented neddylation and activity of cullin E3-ligases, or the lack of deubiquitylase (DUB) activity of this dual-specific enzyme.

### 2.5. Cand1/Tip120A and Cand2/Tip120B

Cullin-associated neddylation-dissociated (Cand; also known as Tip120 (TBP-interacting protein 120)) family members bind to TATA-binding proteins that are the general transcription factors in eukaryotic transcription systems [80]. Although Cand1 is ubiquitously expressed, Cand2 appears to be more muscle specific and upregulated throughout embryogenesis [81]. While Cand proteins are able to interact with transcription factors, their main function is thought to reside in the regulation of CRL activity in cells [12,82]. This is achieved by interacting with CRLs and stabilizing cullin complexes in their inactive form. Cand proteins also modulate substrate-adapter binding as well as prevent modification of cullin proteins by the small ubiquitin-like modified Nedd8 [12,82,83] (Figure 1C). While Cand1 may constitute a more general CRL regulator, Cand2 was specifically shown to regulate cullin-1-containing CRL complexes in muscle cells (Figure 3) [84,85]. The significance of this interaction in muscle is mainly highlighted through cell-based studies. These in vitro analyses revealed that while mostly undetectable in proliferation, Cand2 is strongly induced upon muscle cell differentiation [84]. Cand2 functions may also be regulated by accessory proteins, such as the Hect (homologous to the E6-AP carboxyl terminus) domain E3-ligase Ube3c [86]. Cand2 is an essential protein of muscle differentiation by acting as a negative regulator for the degradation of myogenin and MyoD (myogenic differentiation 1) during myogenesis. More specifically, the interaction of Cand2 with cullin-1 results in a breakdown of the complex containing the linker Skp1 (S-phase kinase-associated protein 1) and the F-box domain-containing substrate adapter protein with myogenin and MyoD (myogenic differentiation 1) [84,85].

Although expressed to a lower amount in cardiac tissues [81], Cand2 has been implicated in genome-wide association study (GWAS) analyses as a candidate gene for atrial fibrillation susceptibility [87,88,89]. Follow-up studies using knockdown of Cand2 in zebrafish models resulted in prolonged action potential duration, further implicating actions of this CRL regulator in cardiac functions [90]. However, targets of Cand2 action in the heart that are causative of the atrial fibrillation phenotype remain unknown. Cand2 interaction with cullin-1, however, suggests the specific deregulation of a substrate for this CRL as a mechanism of action.

## 3. The Role of Cullin-1 Based CRLs in Cross-Striated Muscles

Cullin-1 forms with the RING-domain protein Rbx1 the backbone of the Skp1-cullin-1-F-box (SCF) complex. Due to its overexpression in many cancer forms, it is the most extensively studied member of the cullin family [94,95,96]. Cullin-1-based E3-ligases rely on the small protein Skp1 to interact with substrate adapter proteins of the F-box protein family [7] (Figure 3A). The F-box protein family consists of more than 70 members [7,97], which also feature an array of other protein domains besides the Skp1-binding F-box domain (Figure 3B; Appendix A). *Cullin-1* global knockout mice are embryonic lethal at E6.5 due the inability to degrade cyclin-E [9]. Cullin-1 and its interacting proteins have been mainly associated with various cancers because of their predominant role in cell cycle regulation [95,96]. Hence, various approaches are currently investigated to inhibit all cullin function either directly through Mln4924 [19], via Rbx1-inhibiting proteins like glomulin [56], or through more targeted strategies that selectively suppresses actions of specific cullin substrate adapters, such as Fbxo32/atrogin-1 (Table 1).

Besides F-box proteins Fbxo32/atrogin-1, which is known to be highly expressed in cross-striated muscle tissues, several other F-box-containing proteins that are enriched in cardiac and/or skeletal muscle tissues can be identified (Figure 3C,D) [91,92,93]. Among others, these include the F-box proteins Fbxo40, Fbxo25, Fbxl4, and Fbxw7.

### 3.1. Fbxo32/Atrogin-1/MAFbx

In muscle tissues, the Fbxo32 (also known as Atrogin-1 or MAFbx)-containing SCF-complex counts among the best-studied CRL E3-ligases due to the muscle-specificity of this cullin-1 substrate adapter. Fbxo32-containing CRLs promote muscle atrophy through increased muscle protein degradation by the UPS. Prominent muscle-specific substrates of this E3-ligase include myosin-binding protein C, the transcription factor MyoD, and calcineurin [98,99,100].

Overexpression of Fbxo32 in post-mitotic C2C12 cells led to the formation of thinner myotubes compared to controls, suggesting a role in the balance between muscle fiber atrophy and hypertrophy. Inhibition of Fbxo32 in C2C12 cells triggered an increase in MyoD protein levels, revealing that MyoD is one target of the Skp1-cullin-1-Fbxo32 complex during myoblast differentiation [101].

Fbxo32 is upregulated during muscle development (Figure 3C) and can be highly induced upon muscle immobilization, hindlimb-unloading, or by glucocorticoid treatment [101]. Fbxo32 interacts with α-actinin-2 (Actn2) and localizes at the sarcomeric Z-disc in cardiomyocytes [100]. Fbxo32 is also found expressed in the nucleus of skeletal muscle fibers upon atrophy [102]. Altogether these data highlight the active role of Fbxo32 in muscle mass loss.

Clinically, patients presenting cachexia or muscle wasting due to denervation or immobilization show increased expression of Fbxo32 [103]. Engineered mice lacking Fbxo32 do not show any particular phenotype in normal conditions but display more resistance to muscle mass loss under atrophic conditions [101]. Besides underscoring the importance of Fbxo32 for muscle wasting, this manuscript also highlights partially overlapping functions between the Skp1-cullin-1-Fbxo32 E3-ligase complex and MuRF1/Trim63, an E3-ubiquitin ligase of the muscle ring-finger (MuRF) protein family. Conversely, over-expression of Fbxo32 or MuRF1 in muscles induces muscle atrophy [101].

Lately, studies have been conducted to identify direct and indirect inhibitors of Fbxo32 expression (Table 1), in order to counteract skeletal muscle loss and atrophy in various murine models and patients.

Oligonol, a polyphenol derived from lychee, was found to alleviate muscle atrophy and Fbxo32 overexpression through upregulation of the NADP-dependent deacetylase sirtuin-1 (Sirt1) in db/db animals, a mouse model for obesity and muscle loss [63].

Treatment of immobilized rats with CI-994, an inhibitor of histone deacetylase 1 (Hdac1), prevented increased expression of Fbxo32 in soleus muscle under hindlimb unloading. Administration of this Hdac inhibitor was able to improve skeletal muscle mass after unloading, suggesting that histone acetylation is a major factor in the modulation of muscle atrophy [64].

Roflumilast, a proteolysis inhibitor that targets phosphodiesterase-4, impeded Fbxo32 expression in atrophic myotubes from patients with severe chronic obstructive pulmonary disease [65].

Imperatorin is another promising agent to treat muscle loss in the context of cachexia, as it selectively inhibits the Stat3 (signal transducer and activator of transcription 3)-dependent atrophy-signaling pathway, by downregulating skeletal muscle atrogene expression, such as Fbxo32 [66].

Fbxo32 levels were also regulated by a cellular inhibitor of apoptosis 1 (cIAP1) through an Ikkβ-dependent mechanism. Suppression of NFkB signaling or cIAP inhibition through smac mimetic compounds (SMCs), such as LCL161 (which is a cIAP antagonists that promotes the premature degradation of the proteins), could serve as a potential therapy for skeletal muscle atrophy [67].

Finally, a mutation in *Fbxo32* has been identified in patients with dilated cardiomyopathy. This missense mutation is predicted to impair the binding of Fbxo32 to the Skip1 protein by distorting the fold of the F-box and abrogating the function of the protein [104].

### 3.2. Fbxo40

Similar to Fbxo32, the Fbxo40 protein is postnatally upregulated in skeletal muscles from 2 weeks after birth (Figure 3C). The expression of this adapter protein can be highly induced upon denervation, but not starvation-induced atrophy, and is decreased in limb–girdle muscular dystrophy patients. Besides skeletal muscles, the protein can also be found in hearts without showing any detectable expression in other tissues. As Fbxo40 is not detectable in C2C12 cells, forced expression of a tagged form of Fbxo40 showed a diffuse cytoplasmic localization of the protein [105]. The down-regulation of Fbxo40 in differentiating myoblasts triggered the accumulation of insulin-like growth factor 1 (Irs1) and hyper-phosphorylation of the Akt serine/threonine kinase 1 (Akt, also known as Pkb (protein kinase B)), the main signaling pathway involved in muscle mass growth [106].

Differently from *Fbxo32* knockout mice, *Fbxo40* knockouts exhibit a pronounced muscle phenotype under normal conditions. Knockout mice developed severe hypertrophy, especially during the muscle growth phase, which is sensitive to the activation of the insulin-like growth factor 1 (Igf1)/Akt signaling pathway [106]. On the molecular level, data obtained in *Fbxo40* knockout mice confirmed that Skp1-cullin-1-Fbxo40 targets Irs1 for degradation through the UPS in skeletal muscle, leading to the activation of the Igf1-Akt signaling pathway [106]. These data uncovered the role of cullin-1 in muscle insulin metabolism through Fbxo40 [107]. Surprising results that influence Fbxo40 splicing and possibly function of this cullin-1 substrate adapter came from *Prmt1* (protein arginine methyltransferase 1) knockout mice. Prmt1 catalyzes the methylation of several RNA-binding proteins, leading to alternative splicing and alterations in the genetic program of cells. Cardiac specific *Prmt1* knockouts develop a severe form of dilated cardiomyopathy, accompanied by a previously uncharacterized splice isoform of Fbxo40 [108]. Removal of *Prtm1* resulted in the excision of Fbxo40 exon 3 that encodes for large parts of the protein, leading to a potential loss of proper substrate adapter functionality. However, the exact functions of this splice isoform remain to be discovered.

*Fbxo40* knockout pigs generated via CRISPR/Cas9 technology also developed skeletal muscle hypertrophy without exhibiting any changes to cardiac muscles. Skeletal muscles were characterized by a 4% increase in muscle mass [109]. Similar to the observation made in *Fbxo40* knockout mice, Irs1 levels were increased, coupled with stimulation of the Igf1-Akt hypertrophic signaling pathway [109], confirming the molecular mechanism through which cullin1-Fbxo40 regulates muscle growth.

### 3.3. Fbxl3 and Fbxl21

Two highly homologous substrate adapters of the F-box and leucine-rich repeat protein family that have recently been demonstrated to govern circadian rhythm are Fbxl3 and Fbxl21 [110]. While both proteins target Cry (cryptochrome) for degradation through formation of a cullin-1-based E3-ligase, loss of function for Fbxl3 lengthens the periodicity of the cellular clock, whereas mutations in *Fbxl21* cause shortening of the circadian rhythmicity. Besides the choice of Fbxl3 or Fbxl21 as substrate adapter, the authors also demonstrated that subcellular compartmentalization of the two cullin-1 complexes in the nucleus or cytoplasm of the cell plays vital roles for the modulation of Cry expression. In muscles, Fbxl21 also modulates protein levels of Tcap (titin cap, also known as telethonin), a Z-disc protein that forms a trimeric complex with the N-termini of two titin proteins [111]. Fbxl3 also binds to Tcap but does not modulate its protein levels. Loss of Fbxl21 functionality in mutant skeletal muscles disrupted the circadian oscillation of Tcap in a Gsk-3β-dependent manner, leading to smaller fiber sizes and reduced skeletal and cardiac muscle function [112].

### 3.4. Fbxl4

F-box and leucine-rich repeat protein 4 (Fbxl4) is an orphan F-box protein localized to mitochondria [113]. Mutations in *Fbxl4* have been reported in patients with early-onset encephalomyopathic mitochondrial DNA depletion [114,115,116]. Clinical symptoms include impairment of fetal movements and severe hypotonia at birth [114,115]. Analysis of muscle biopsies revealed a decrease in mitochondrial mass as well as hyper fragmentation, respiratory chain deficiency, and a loss of mitochondrial membrane potential [113,114].

Recent studies revealed that a mutation in *Fbxl4* is also associated with cardiac hypertrophy [117]. In-depth characterization showed that Fbxl4 acts as a regulator of mitochondrial fusion [117], which likely explains the mitochondrial defects observed in patients [117,118]. Intriguingly, administration of dichloroacetate (DCA), a mitochondria-targeting small molecule that alters mitochondrial metabolism [119], improved clinical symptoms in one patient and reversed the cardiac hypertrophy [117].

### 3.5. Fbxl10/Kdm2b

The role of F-box protein 10 (Fbxl10; also known as lysine demethylase 2b (Kdm2b)) in muscle tissues is poorly understood. Fbxl10 functions as a demethylase for histones and is a component of the noncanonical polycomb repressive complex 1 [120]. However, Fbxl10 expression is lowered in the heart of diabetic rats compared to healthy control hearts [121]. Overexpression of Fbxl10 in cardiac tissue protects from diabetes-related cardiac dysfunction, cell death, inflammation, and oxidative damage through the modulation of the protein kinase Cβ (Pkcβ2) signaling pathway [121].

Fbxl10 expression reduced myocardial infarction, remodeling, and inflammatory response in myocardial ischemia-reperfusion rats by suppressing the expression of endoplasmic reticulum stress proteins including DNA damage-inducible transcript 3 (Ddit3, also known as Chop), heat shock protein family A member 5 (Hspa5, also known as Grp78), activating transcription factor 4 (Atf4, also known as Creb2), and phospho-Erk (mitogen-activated protein kinase) [122].

### 3.6. Fbxl16

F-box and leucine-rich repeat protein 16 (Fbxl16) is different from most F-box proteins. In addition to interacting with Skp1 and forming an SCF complex [123], it also binds independently to protein phosphatase 2A (PP2A) to negatively regulate phosphorylation of vimentin [124]. This function of Fbxl16 is thought to be important for the differentiation of stem cells along the Flk1+ lineage, which form cellular precursors for cardiac tissue formation.

### 3.7. Fbxl22

F-box and leucine-rich repeat protein 22 (Fbxl22) is enriched in heart muscle and localizes to the sarcomeric Z-disk [125]. In vitro, Fbxl22 interacts with α-actinin-2 (Actn2) and filamin-C, two sarcomeric proteins that are important for muscle contraction and sarcomeric structure. Fbxl22-containing CRLs mediate degradation of these sarcomeric proteins through their poly-ubiquitylation before clearance via the proteasome [125].

The knockdown of *Fbxl22* in zebrafish leads to a marked accumulation of Actn2 and a progressive reduction of cardiac contractility, confirming the importance of the Skp1-cullin-1-Fbxl22 complex for cardiac muscle function in vivo [125].

Recently, the temporal and spatial regulation of murine Fbxl22 was elucidated by Hughes and co-workers for skeletal muscles [126]. Intriguingly, the authors uncovered a novel Fbxl22 splice isoform in muscles. Both SCF substrate adapter splice isoforms are induced upon muscle cell differentiation and became further upregulated during neurogenic skeletal muscle atrophy. Overexpression of Fbxl22 in vivo resulted in a transient increase in muscle mass that was accompanied by cell-infiltration, necrosis, muscle degeneration, and increased occurrence of centralized nuclei. At the molecular level, Fbxl22 overexpression affected protein levels of dystrophin, desmin, vimentin, and sarcomeric Actn2. However, alterations in Actn2 levels were Fbxl22 isoform-dependent, displaying a significant reduction of Actn2 only upon loss of the shorter Fbxl22 transcript. Molecular analyses also revealed evidence for altered autophagy and UPS-mediated protein turnover. Intriguingly, the ubiquitylation profile of muscle proteins was again only affected in one of the two splice isoforms. The authors also performed loss of function experiments and found sparing of muscle fibers during denervation induced atrophy in wildtype and *MuRF1* knockout muscles.

Despite these advances, Fbxl22 function in human skeletal myopathy and cardiac pathology has not been explored. Combined, these data add Fbxl22 as another cullin-1 substrate adapter that is important for muscle tissue development, function, and maintenance, specifically during denervation-induced atrophy.

### 3.8. Fbxo38

F-box protein 38 (Fbxo38) is expressed in the spinal cord, muscles, and the brain [127]. Fbxo38 is a poorly investigated F-box protein but was shown to function independently from the SCF complex [128]. Fbxo38 is a co-activator of the transcription factor Krüppel-like factor 7 (Klf7) that regulates neurogenesis and cell cycle progression [129]. Missense mutation of *Fbxo38* in patients leads to distal spinal muscular atrophy with calf predominance [127]. Studies of the pathological mechanism at stake in this disease point towards an impaired Fbxo38-Klf7 complex and deregulation in the expression of Klf7 target genes, including cyclin-dependent kinase inhibitor 1 (Dkn1A), neurotrophic receptor tyrosine kinase 1 (Ntrk1), and L1 cell adhesion molecule (L1CAM), which are important for axonal development and motor neuron maintenance [127].

### 3.9. Fbxo25

F-box protein 25 (Fbxo25) is cardiac-specific, and its expression is highly dynamic over cardiogenesis. This adapter protein is known to interact with the Skp1-cullin-1 complex to degrade cardiac transcription factors [130]. Indeed, Fbxo25 facilitates the degradation of Nkx2.5 (Nk2 homeobox 5), Isl1 (insulin gene enhancer protein Isl-1, also known as islet-1), Hand1 (heart and neural crest derivatives expressed 1), Tbx5 (T-box transcription factor 5), and Mef2C (myocyte enhancer factor 2C), suggesting a pivotal function for muscle protein homeostasis and cardiac development [130,131].

### 3.10. β-Trcp1 (Btrc/Fbxw1A) and β-Trcp2 (Btrc2/Fbxw11)

The β-transducin repeat-containing proteins β-Trcp1 (Fbxw1A) and β-Trcp2 (Fbxw11) have been intensively investigated in smooth muscles [132,133]. A recent study using the proximity-dependent biotin identification (BioID) technology revealed that β-Trcp1/2-SCF complexes target at least 50 proteins for degradation through the UPS [134]. In cardiac cells, β-Trcps are involved in the regulation of the insulin signaling pathway through degradation of Phlpp-1 (PH domain leucine-rich repeat protein phosphatase-1) [135]. Phlpp-1 dephosphorylates and inactivates Akt, a regulator of the survival signal pathway [136]. Mechanistically, insulin enhances the binding between Phlpp-1 and β-Trcp, targeting the enzyme for ubiquitin-dependent degradation [135].

### 3.11. Fbxw7

Fbxw7 is another F-box protein that forms a Skp1-cullin-1-F-box complex with high expression in skeletal muscles (Figure 3C,D). It is a well-known tumor suppressor that degrades oncoproteins such as cyclin-E, notch receptor 1 (Notch1), c-Jun (Jun proto-oncogene, AP-1 transcription factor subunit), c-Myc (Myc proto-oncogene), or the mammalian target of rapamycin kinase (mTOR) [137,138,139,140,141]. Mutations in *Fbxw7* have been identified as a driver of cancer development [142,143], but more recently it has been also shown to act as a regulator of normal tissue growth [144,145].

Fbxw7 encodes three isoforms (α, β, and γ) through alternative splicing, which in addition show specific expression patterns [146]. Fbxw7β and Fbxw7γ are dynamically regulated during myogenesis. Indeed, Fbxw7β was downregulated during myoblast differentiation, while Fbxw7γ expression was strongly upregulated [147]. In vitro, Fbxw7β overexpressing myoblasts were unable to properly differentiate into myotubes. The authors of this study found that the Fbxw7β isoform negatively regulates primary myoblast differentiation, proliferation, and migration of muscle cells and satellite cells, revealing the essential role of this E3-ubiquitin ligase for skeletal muscle regeneration [147].

In line with in vitro data, mice overexpressing the β-isoform of Fbxw7 undergo skeletal muscle atrophy, which is mediated by the up-regulation of major atrogene markers such as Fbxo32 and MuRF-1 [148]. These results point out the importance of the developmental down-regulation of Fbxw7β for proper myogenesis and muscle growth [147].

### 3.12. Other F-Box Proteins

The literature provides several F-box proteins that are highly expressed in skeletal and cardiac muscle tissues, including Fbxo17, Fbxo27, Fbxw2, and Fbxw5 (Figure 3C,D). However, their molecular roles in muscle development or function remains unknown or poorly characterized.

F-box only proteins Fbxo17 (Fbg4), Fbxo44 (Fbg3), Fbxo27 (Fbg5), and Fbxo16 are highly expressed in various tissues, including cardiac and/or skeletal muscles [149,150]. F-box and WD repeat domain-containing proteins Fbxw2, Fbxw5, and Fbxw6 are also present in multiple tissues, including the heart and skeletal muscle tissues (Figure 3C) [151,152].

Functions of Fbxo16, Fbxo17, and Fbxw2 have all been linked to the regulation of the Wnt/β-catenin signaling pathway in non-muscle cells [153,154,155,156,157,158]. Fbxo27 was shown to direct damaged glycoproteins on the surface of lysosomes, including Lamp1 and Lamp2, towards autophagy-mediated degradation [159].

Fbxo44 was shown to mediate the degradation of Rgs2. Intriguingly, this function for Fbxo44 is linked to actions of cullin-4, rather than a cullin-1-based E3-ligase complex [160]. Another F-box domain protein whose functions are linked to either cullin-1 [161] or cullin-4 [162]-based E3-ligase complexes is Fbxw5. Functions of this substrate adapter have been linked to the regulation of autophagy [163,164], and it recently emerged on the list of deregulated mRNAs in a novel transgenic mouse model for heart failure and cachexia [165].

Fbxo31 is developmentally upregulated in skeletal muscles (Figure 3C) and displays fiber-type specificity in adult muscles (Figure 3D). This F-box-containing protein was linked together with Fbxo21 (also called Smart (specific of muscle atrophy and regulated by transcription)) to a novel biomechanical signaling pathway, as their expression is induced by mechanical silencing [166]. However, this finding is contested, as another group detected no changes to their expression during passive mechanical unloading caused by denervation [167]. Active Akt and removal of forkhead box (Fox)O transcription activity by muscle-specific knockout of *FoxO1*, *FoxO3*, and *FoxO4* prevented Fbxo31 and Fbxo21 upregulation [166,168]. All three FoxO transcription factors are downstream of Akt, which also regulates muscle-specific atrogens Fbxo32 and MuRF1. Intriguingly, Fbxo21 was also identified among the few cullin-adapter proteins with sex-specific expression profiles in hearts [169].

Less well-characterized is also the role of Fbxo2 (also known as Ocp1) for cross-striated muscles. This F-box adapter is suspected to regulate glucose homeostasis through ubiquitylation of the insulin receptor (InsR) in obese mice [170], although this result is contested [171,172]. A possible function for Fbxo2 may be found in regulating connexin channel proteins. It is known to associate with connexin-26 (also known as gap junction protein β2 (Gjb2)) [173] and is found downregulated in *connexin-43* (gap junction protein α1 (Gja1)) knockout mice [174].

Further analysis will be required in order to decipher the functions of these and other F-box-containing proteins for cross-striated muscles.

## 4. Functions of Cullin-2 in Cardiac and Skeletal Muscles

Cullin-2 is ubiquitously expressed and interacts with elongin-B and elongin-C as well as an array of substrate adapter proteins to form a functional cullin-RING ligase (Figure 4A) [175]. Like all cullin E3-ligases, cullin-2 is involved in the stability and degradation of various proteins, regulating a multitude of biological processes (e.g., germline differentiation) and participating in the pathology of diseases, such as cancer or viral defense [175]. The tumor suppressor protein von Hippel–Lindau (VHL) represents the most well-known substrate adapter for the cullin-2-based E3-ligase [175]. Binding of VHL and other cullin-2 adapter proteins to elongin-B/C depends on the presence of the VHL-box motif (sometimes also referred to as the B/C-box with adjacent Cul2-box motif) (Figure 4A). The short 10 amino acid sequence of the B/C-box is characterized by the degenerate consensus [STP] LXXX [CSA] XXXΦ, with leucine in position two emerging as a key residue for docking to a hydrophobic binding pocket in the elongin heterodimeric linker [176,177,178]. There are similarities of the VHL-box with the 40 amino acid Socs-box motif, characteristic of cullin-5 substrate adapter proteins that also utilize the elongin-B/C heterodimer for the formation of the functioning E3-ligase complex. Adjacent to the B/C-box is the cullin-2 box, which further defines the specificity of substrate adapter proteins for this E3-ligase over cullin-5-based E3-ligases. Both motifs (the B/C-box and the Cul2-box) are separated by between 3 and ~80 amino acids [179]. However, compared to the Cul5-box with its relatively conserved LPΦP sequence motif, the Cul2-box is far more diverse in its primary sequence.

Recently, the structural basis for the selectivity of elongin-B/C substrate adapter linker proteins towards cullin-2 or cullin-5 has been further investigated [180]. The authors conclude that sometimes quite subtle sequence differences in cullin-2 vs. cullin-5 themselves, as well as the corresponding B/C-box and Cul2- vs. Cul5-box motifs provide the structural basis for the preferential association of these E3-ligases to their substrate adapters. Of note, the complex of the elongin-B/C heterodimer with a substrate adapter forms prior to the recruitment of either cullin-2-Rbx1 or cullin-5-Rbx2.

Functional activity of cullin-2-based E3-ligase complexes can be inhibited by the divalent metal ions cobalt and nickel (Table 1), which bind to three different regions in the cullin protein. Interestingly, the association of Co^2+^ ions does not abrogate the binding of elongin-B/C or the VHL substrate adapter, suggesting an allosteric inhibition of the E3-ligase activity [57]. A more recent approach to modulating cullin-2 activity is based on short peptide sequences that disrupt the binding of cullin-2 to the elongin-B/C heterodimer (Table 1) [58]. However, this approach is currently marred by challenges, such as low affinity or the complex structural nature of the protein interaction network required to form cullin-2-based E3-ligases. 

Several proteins have been identified that serve as bona fide substrate adapters for cullin-2-elongin-B/C-based E3-ligase complexes (Figure 4B,C). Some of these adapters were found based on sequence comparison with the known cullin-2 substrate adapter VHL, while a newer study used affinity purification of elongin-B-associated proteins followed by mass-spectrometry to identify potential cullin-2 adapters and substrates [179].

Few data are currently available on the role of cullin-2 and its substrate adapters in cross-striated muscle cells. However, cullin-2 expression is induced following artificial exercise in C2C12 cells, suggesting a role in the structure of the sarcomere and especially in the thin filament architecture that is remodeled after the termination of the exercise [182].

Cullin-2 mutants and knockdown flies show defects at the larval neuromuscular junction (NMJ), including impaired number of synaptic boutons on muscles. Indeed, mutations in both *cullin-2* and *cullin-5* result in an increase of bouton numbers in motoneurons innervating the NMJ [183]. Further studies are required to identify cullin-2 substrates that are involved in cross-striated muscle development, function, or NMJ development.

Some of the bona fide and potential substrate adapters for cullin-2 show muscle-specific expression (Figure 4B,C; Appendix A) [91,92,93]. These include proteins of the Kelch domain-containing (Klhdc) protein family, the von Hippel-Lindau tumor suppressor protein VHL, or proteins of the fem-1 homolog (Fem1) protein family.

### 4.1. VHL Tumor Suppressor Protein

The cullin-2 elongin-B/C substrate adapter VHL (von Hippel–Lindau tumor suppressor, also known as pVHL) is best known for its association with the syndrome of the same name [184]. The syndrome is known to affect multiple organs and may also result in cardiovascular disease in affected patients.

The best-characterized substrate for VHL is the hypoxia-inducible factor 1α (Hif1-α) [185]. Regulation of Hif1-α by the VHL-containing cullin-2 E3-ligase is critically involved in the metabolic switch during cardiac maturation, between embryonic and adult stages [186]. Up until the mid-gestation of cardiac development, the muscle is compartmentalized, with a glycolytic metabolism in the compact myocardium and an oxidative one in the trabeculae. After mid-gestation, the compact myocardium switches to an oxidative metabolism [186,187].

Following this pattern, the transcription factor Hif1-α is highly expressed in the compact myocardium until mid-gestation, when it becomes down-regulated through VHL-mediated poly-ubiquitylation and subsequent degradation by the UPS [186].

The absence of VHL in cardiomyocytes leads to degeneration, malignant transformation, and failure of the heart [188], confirming the essential role of the cullin-2-elonginB/C-VHL complex for the regulation of Hif1-α and its effects on cardiac physiology and development [189]. Besides Hif1-α, recent data suggest that the cardiac-specific phospholamban (Pln) protein represents another substrate for the VHL-containing cullin-2 E3-ligase [190]. Pln is a well-characterized regulator of the sarco/endoplasmic reticulum Ca^2+^-ATPase (SERCA), thereby modulating cardiac calcium handling and contractility [191,192]. Intriguingly, divalent Co^2+^ ions, a potent cullin-2 inhibitor of Hif1-α degradation, did not affect Pln levels [190].

In cardiac endothelial cells, miR-424, a miRNA induced upon hypoxia, directly targets the expression of cullin-2 (Table 1) [59]. In the context of hypoxia, down-regulation of cullin-2 through this mechanism leads to the stabilization of the transcription factor Hif1-α and promotes angiogenesis, a critical adaptation for cell survival [59].

In skeletal muscle, the VHL-regulated Hif1-α signaling pathway contributes to the normal physiological muscle response following exercise [193], including modulation of muscle metabolism, oxygen consumption, and anaerobic glycolysis [194]. Intriguingly, this response is dependent on Bmal-1 (brain and muscle Arnt-like 1), a key circadian rhythm regulator, suggesting differential activity of this pathway dependent on the solar cycle [194].

### 4.2. Cullin-2 Substrate Adapters of the Fem1 (Fem-1 Homolog) Family

Fem1 proteins, a family of evolutionary-conserved VHL-box adapter proteins for cullin-2-elongin-B/C E3-ligases, are upregulated during muscle development [195] and highly expressed in mature skeletal muscles and heart (Figure 4B) [196]. Biochemical and microscopy analyses in muscle cells showed that Fem1a localizes within the mitochondria and is up-regulated in the heart upon ischemia-reperfusion injury [197]. Fem1 proteins are also increased in patients with Keshan disease, a subtype of cardiomyopathy [198]. The association of Fem1 proteins with this disease indicates that these cullin-2 adapter proteins play important roles for cardiac homeostasis. Substrates for the cullin2-elongin-B/C-Fem1 complex have not been investigated specifically in muscle cells, but Fem1 adapters have been shown to regulate stem-loop binding proteins, which have important functions for mRNA cleavage, translation, and degradation [199]. In addition, many Fem1 substrates contain C-terminal glycine-ended degron motifs, which follow the “C-end rule” pathway that is proposed to widely govern protein stability [200].

### 4.3. Other Cullin-2 Elongin B/C Substrate Adapter Proteins

Klhdc1 and Klhdc2 are Kelch repeat proteins that are both highly expressed in skeletal muscles. While Klhdc2 is predominantly localized to the nucleus, Klhdc1 is a cytoplasmic protein [201].

Klhdc proteins serve as substrate adapters for cullin-2-based E3-ligase complexes [202]. Similarly to Fem1 proteins, Klhdc2/3 recognizes c-terminal glycine-ended degrons in a large variety of proteins [200], suggesting that these cullin-2 E3-ligases target many substrates within muscle cells.

Another cullin-2 elongin-B/C substrate adapter protein that is highly expressed in skeletal muscles is the cell cycle regulator zyg-11 family member B (Zyg11b) (Figure 4B,C) [203]. Currently, muscle-specific roles for Zyg11b have not been explored. However, ENU (N-ethyl-N-nitrosourea) mutant mice that exhibit truncations in the region of Zyg11b show perinatal lethality and cardiac defects [204].

## 5. The Roles of Cullin-3 E3-Ligase Complexes in Cardiac and Skeletal Muscles

Like most of cullins, cullin-3 is ubiquitously expressed. However, its emerging role in skeletal muscles has recently been described and investigated [205,206]. Cullin-3 associates in its N-terminus with adapter proteins that contain a BTB (BR-C, ttk, and bab) domain, which is also known as a POZ (Pox virus and zinc finger) domain (Figure 5A). Cullin-3 localizes to the region of the sarcomeric Z-disk in adult heart and skeletal muscle tissues [207], perhaps by binding to Capz [208], or via a direct or indirect interaction with (an) as of yet unknown Z-disk protein(s) (Figure 5B). However, this pattern is developmentally regulated, as cullin-3 localization in early embryonic heart at E15.5 is mostly restricted to diffuse speckles in the cytoplasm and nucleus of cardiac myocytes (Figure 5C, top panel). Later in embryonic development at E18.5, the localization becomes restricted to the nucleus, presumably playing part in cardiomyocyte proliferation and differentiation, and the sarcomeric Z-disk (Figure 5C, bottom panel). Cullin-3 is also found at the intercalated disk, such as in *obscurin* knockout hearts [207], indicating that the strictly sarcomeric localization of this cullin protein in adult tissues may be subject to cardiac stresses. The sarcomeric localization of cullin-3 indicates its involvement in the degradation (via poly-ubiquitylation) [207] or stabilization (via mono-ubiquitylation) [209] of myofilament substrate proteins. The exact roles for cullin-3 at each developmental step in cardiac and skeletal muscles are determined by the muscle-specific subset of substrate adapter proteins, which exhibit surprisingly diverse expression and localization patterns as well as functions within cross-striated muscles.

Cullin-3 substrate adapters typically contain BTB-domains, which link the cellular substrate to the cullin-3 N-terminus. Of the more than 180 BTB-domain-containing proteins in the human genome (Figure 5D; Appendix A), several have been shown to play important roles in skeletal muscle development or disease, such as Klhl40, Klhl41, and Kbtbd13.

However, publicly available RNAseq and protein-expression datasets suggest that many more BTB-domain proteins may play crucial roles in cardiac and skeletal muscle development, function, and disease. Indeed, expression analysis of BTB-domain proteins indicates that many cullin-3 substrate adapters show characteristic temporal regulation of their expression levels over the time-course of normal heart or skeletal muscle development (Figure 5E) [91,92]. In addition, we and others [93,206,211] have found that cullin-3 substrate adapters of the BTB-domain family display an extraordinary spatial diversity among skeletal muscle tissues (Figure 5F,G). A recent publication from Terry and co-authors [211] that investigated 11 skeletal muscles as well as smooth and cardiac muscles from rodents identified Klhl34 as contributing to one of the most statistically significant co-expression analyses modules characterizing soleus muscles, while Klhl29 was more characteristic of flexor digitorum brevis (FDB) muscles, and tongue muscles were enriched in Kctd1, Kctd4, Kctd5, Zbtb7c, Btbd11, and Btbd19. Additional cluster analysis revealed that Zbtb18, Zbtb44, Kbtb12, Kbtbd13, Rhobtb3, and Btbd6 were more associated with fast-twitch muscle types. While transcriptome-data are suggestive of differences in muscle-type expression of cullin-3 adapters, expression profiling using antibody data appear to give more pronounced results. Indeed, we found that diaphragm muscles represent a hotspot for the expression of several cullin-3 substrate adapters, such as Kctd6, Klhl9, and Zbtbd16, while other BTB-domain-containing adapters, such as Klhl41 and Kbtbd13, appear to be more ubiquitously expressed among many muscle-types (Figure 5G) [206].

However, it is important to keep in mind that the presence of a BTB-domain does not necessarily guarantee the functionality of a BTB-domain-containing protein as cullin-3 substrate adapters. Our data and those from other groups suggest that some BTB-proteins are not participating in the formation of cullin-3 E3-ligase complexes [20,212,213].

In vitro, the neddylated form of cullin-3 is upregulated during both early C2C12 cell differentiation and AChR clustering following agrin stimulation [20], suggesting important functions for this specific cullin E3-ligase throughout myogenesis.

*Cullin-3* global knockouts are embryonic lethal around E7.5, due to failure in cyclin-E degradation [8]. In order to study the function(s) of cullin-3 specifically in skeletal and cardiac muscles, conditional knockout animals have been generated using the myogenin-cre and α-myosin heavy chain (αMHC)-Cre promoter, respectively [205,206]. Both muscle-specific knockouts exhibit neonatal lethality due to non-functional skeletal muscles or a severe cardiomyopathy, respectively. Loss of cullin-3 in skeletal muscles was characterized by underdeveloped muscles, the formation of protein aggregates visible in Gomori trichrome stains, and severely disrupted neuromuscular endplate development [205,206]. Skeletal muscles also exhibited pathological accumulation of non-muscle actinin isoforms (Actn1 and Actn4) as well as deregulation of other thin-filament-associated proteins that have been linked to the development of nemaline myopathy. Cardiac-specific knockout of *cullin-3* leads also to protein aggregate formation in cells that is accompanied by vacuoles, as well as a severely altered metabolic profile [205].

While the specific roles for many of the potential BTB-domain-containing substrate adapters with characteristic expression profiles in muscles remain to be investigated (Figure 5E–G), several cullin-3 adapter proteins have been shown to play crucial roles in muscle development or function.

### 5.1. Klhl40/Kbtbd5

Kelch-like protein 40 (Klhl40)/Kelch repeat and BTB-domain-containing protein 5 (Kbtbd5) is restricted to differentiating muscle cells and acts as a cullin-3 substrate adapter [214]. During myogenic differentiation, Klhl40 promotes the poly-ubiquitylation and degradation of DP1, thereby inhibiting the activity of the E2F1-DP1 complex, a pro-apoptotic transcription factor [215]. Consequently, *Klhl40* knockout mice develop atrophic and disorganized skeletal muscles and increase expression of apoptotic genes such as Bnip3 (Bcl2 interacting protein 3) and Trp53inp1 (transformation related protein 53 inducible nuclear protein 1). Eventually, knockout animals die within 3 weeks after birth due to strong muscle weakness [215]. In skeletal muscles, Klhl40 localizes to the sarcomeric I-band and A-band by binding to the thin-filament-associated nebulin protein [209].

Unexpectedly, cullin-3-Klhl40 linked E3-ligase complexes do not only lead to the poly-ubiquitylation and subsequent degradation of muscle substrates, but they may stabilize substrate proteins through mono-ubiquitylation. Leiomodin-3 (Lmod3), a tropomodulin-related protein that promotes actin nucleation, was suggested to be stabilized through this mechanism [209]. Consequently, loss of Klhl40 in muscles leads to the destabilization of thin filament proteins including Leiomodin-3.

In humans, mutations in the *Klhl40* gene have been associated with various forms of nemaline myopathies [216,217,218,219,220,221,222,223,224]. Intriguingly, acetylcholinesterase inhibition initiated a sustained positive response in a patient with a *Klhl40* mutation, suggesting a role for this BTB-domain-containing protein in NMJ formation and function [225].

### 5.2. Klhl41/Kbtbd10

Similar to Klhl40, Kelch-like family member protein 41 (Klhl41; also known as Kbtbd10, Krp1, or sarcosin) is a muscle-specific cullin-3 substrate adapter protein [226,227]. In humans, mutations in *Klhl41* are also associated with nemaline myopathy development [228].

Molecular analysis in zebrafish shows that knockdown of *Klhl41* causes nemaline myopathy-like abnormalities, characterized by aberrant myofibril formation [228]. In mice, knockout of *Klhl41* leads to neonatal lethality within 12 days after birth [227]. Similar to the zebrafish model, knockout mice develop a severe nemaline myopathy phenotype characterized by protein aggregates, nemaline bodies, sarcomere disarray, and Z-disk streaming [227]. At the molecular level, Klhl41 is thought to be required for the stabilization of several sarcomeric proteins, including the thin-filament interacting protein nebulin. Surprisingly, sarcomeric substrate proteins of the Klhl41-containing cullin-3 E3-ligase complex are suggested to be poly-ubiquitylated, which greatly increases their half-life within skeletal muscles. Indeed, Klhl41 mediated poly-ubiquitylation is thought to play the role of a chaperone for nebulin, impeding its premature degradation by other E3-ligases [227].

### 5.3. Kbtbd13

Kelch repeat and BTB-domain-containing 13 (Kbtbd13) is primarily expressed in striated muscles and lungs [229]. Kbtbd13 is a substrate adapter for cullin-3 and forms a functional ubiquitin E3-ligase [230]. Mutations in *Kbtbd13* are associated with the NEM6 form of nemaline myopathy, which is characterized by the appearance of core-rods in electron microscopic images [229,231,232]. Our recent study suggests that non-muscle α-actinin-1 (Actn1) and 4 (Actn4) are accumulated in both muscles of *Kbtbd13* patients and *cullin-3* knockout mice [206]. Moreover, forced overexpression of non-muscle Actn1 in vitro inhibits the differentiation and fusion of myoblasts, suggesting a new pathogenic mechanism in cullin-3 and Kbtbd13 related muscle diseases [206].

### 5.4. Kctd6

Potassium channel tetramerization domain-containing 6 (Kctd6) is a substrate adapter for cullin-3 that regulates levels of small ankyrin-1 isoform 5 (sAnk1.5) in cardiac and skeletal muscle cells. Interaction of sAnk1.5 with obscurin, a giant sarcomeric protein prevents the activation of this cullin-3 dependent degradation mechanism in healthy wild-type cells [207]. The mechanism of action is thought to depend on the sequestration of sAnk1.5 via obscurin at the sarcomeric M-band, 1 µm away from the localization of the sAnk1.5 substrate adapter Kctd6 and cullin-3 that localize to the sarcomeric Z-disk. Loss of obscurin results in mislocalization of sAnk1.5 to the Z-disk, subsequent complex formation with the cullin-3-Kctd6 E3-ligase complex, and premature degradation of sAnk1.5 [207]. In combination with Kctd11, Kctd6 is also involved in the degradation of histone deacetylase 1 (Hdac1) [233], a well-known regulator of cardiac morphogenesis, growth, and contractility [234].

### 5.5. Keap1/Klhl19

Kelch-like ECH-associated protein 1 (Keap1, also known as Klhl19) is a substrate adapter for cullin-3 that regulates the levels of Nrf2 [235], which is important for skeletal muscle glycogen metabolism, oxidative stress, and cardiac protection [236,237]. In basal conditions, Keap1 sequesters Nrf2 (erythroid 2-like 2 nuclear factor, also known as Nfe2l2) in the cytoplasm where it is marked for degradation [238]. Upon oxidative stress, Keap1 translocates to the nucleus where it heterodimerizes with transcription factors of the musculoaponeurotic fibrosarcoma (Maf) protein family to reduce stress on striated muscle tissues [238].

### 5.6. Rhobtb3

Rhobtb3 is an atypical member of the Rho family of small GTPases, which is highly expressed in the brain, uterus, and heart [239] (Figure 5E). The only functional data available for Rhobtb3 come from the characterization of a gene trap mouse strain. Mice lacking Rhobtb3 had reduced viability, were smaller, and displayed increases in normalized heart weights [239]. Loss of Rhobtb3 in hearts only led to modest deregulation of the cardiac gene program. Highly deregulated genes included cGMP-specific phosphodiesterase 5A (Pde5a), mitogen-activated protein kinase kinase 4 (Map2k4), and cadherin 2 (Cdh2). Further characterization is necessary to understand the functions of Rhobtb3 in muscle tissues.

### 5.7. Klhl26

Mutations in Klhl26 (Kelch-like family member protein 26) have been associated with Ebstein’s anomaly. Patients manifest with left ventricular non-compaction [240]. In silico analysis suggests that this variant may disrupt the interaction with cullin-3, but functional analysis will be required to confirm its pathogenicity [240]. Substrates of Klhl26 in cardiac cells are still unknown and muscle-specific functions for this cullin-3 substrate adapter protein await further characterization.

### 5.8. Other BTB-Domain-Containing Proteins

Btbd1 is preferentially expressed in cardiac and skeletal muscles [241] (Figure 5E). Expression of its C-terminus inhibited C2C12 differentiation, suggesting important functions for the development of skeletal muscles [242,243]. On the molecular level, Btbd1, together with its close relative Btbd2, were found to interact with topoisomerase-1 [243,244]. Loss of Btbd1 regulation over topoisomerase-1 reduced the nuclear content of this DNA-modifying enzyme in C2C12 cells [243]. Both Btbd1 and Btbd2 show colocalization with Trim5δ in cytoplasmic bodies. Association of the BTB-domain proteins with the RBCC/tripartite motif protein Trim5δ, its ubiquitylation, and subsequent degradation was mediated through the RING-domain of this E3-ligase [245]. Intriguingly, Btbd1 was also among the few cullin E3-ligase substrate adapters that show a sex-specific expression pattern in hearts [169].

Kctd12 is another putative substrate adapter protein that is developmentally regulated in the heart and skeletal muscles (Figure 5E). While little is known about its subcellular localization, a report of its modulation of Gaba(B) receptor functions may have implications for Kctd12 function in muscle diseases that affect spasticity, such as dystonia or multiple sclerosis [246]. Structurally, the BTB-domain in Kctd12 was suggested to bind to the cytoplasmic tail of Gaba(B) receptors. Upon receptor activation, Kctd12 associates with Girk-channels bound to small G-proteins (Gβγ), which become dissociated from this class of potassium channels, leading to their desensitization [247]. Intriguingly, this report suggests non-canonical functions of Kctd12 that are independent of a putative role in cullin-mediated protein degradation. Indeed, Kctd12 counts among the few BTB-domain adapter proteins that do not seem to interact with cullin-3 [212].

## 6. Cullin-4A/4B Based E3-Ligases and Their Function for Cross-Striated Muscles

Cullin-4A and cullin-4B share a high degree of sequence similarity with the exception of the extended N-terminus in cullin-4B [248]. Cullin-4A is highly expressed in the testis, heart, and skeletal muscles [249], while cullin-4B is most highly expressed in the pancreas, endocrine glands, the cerebellum, the lower gastrointestinal tract, bone marrow, and the testes [250]. Due to their structure homology, cullin-4A and cullin-4B have overlapping functions. However, due to their different expression patterns, they also exhibit distinct functions and regulate different substrates [248].

In humans, mutations or deregulation of cullin-4A is mostly linked to the development of cancers (hepatocellular carcinoma and breast cancer) [251].

Mutations in *cullin-4B* are linked to the development of X-linked mental retardation syndrome, associated with aggressive outbursts, seizures, relative macrocephaly, central obesity, hypogonadism, pes cavus, and tremors [252]. These clinical phenotypes confirm an important role for the cullin-4B E3-ligase in neuronal function and development [253].

The cardiac and skeletal muscles of patients with mutations in *cullin-4A* and/or *cullin-4B* have not been studied in detail and will require further analysis. However, *cullin-4A* knockout mice develop a cardiac phenotype characterized by increased hypertension, weakened heart function, and cardiac hypertrophy in male mice only [68,254]. Intriguingly, the phenotype in male mice can be partially rescued by deletion of one *Grk2* (G protein-coupled receptor kinase 2) allele, suggesting a role of cullin-4 in the proper regulation of G-protein coupled receptor (GPCR) signaling pathways [68].

Cullin-4 E3-ligases bind to Ddb1 (DNA damage-binding protein 1), which in turn acts as a linker protein to an array of more than 100 validated or suspected adapter proteins (Appendix A). Many cullin-4 adapters contain at least one DWD-box motif (Ddb1-binding and WD40 repeat; also called the cullin-4–Ddb1-associated WD40 repeat (CDW) or WDXR-motif) [255,256] (Figure 6A, top panel). Although the DWD-box motif in Dcaf proteins (Ddb1 and Cul4-associated factors) was shown to associate with Ddb1 [257], other residues, motifs (such as the F/YxxF/Y motif, or the viral H-box motifs), and protein domains may participate in the interaction of substrate adapter proteins with cullin-4A/B-Ddb1. An example of the large variety in cullin-4 substrate adapters that do not contain a classical DWD-box motif is the heterodimer of Det1 and Dda1, which forms together with the E2-enzyme Ube2e a functional substrate adapter for the cullin-4-Ddb1 E3-ligase [258] (Figure 6A, bottom panel).

*Ddb1* knockouts are embryonically lethal before E12.5 [259]. Tissue-specific knockouts for the central nervous system indicate that Ddb1 severely affects the proliferation of progenitor cells, leading to massive cell death. However, no muscle-specific knockout for *Ddb1* has been reported. Intriguingly, functions of Ddb1 as an intermediary between cullin-4 and substrate adapters of the Gβ protein family can be modulated by phosphorylation at Ser645 via cAMP dependent protein kinase (Pka) in an evolutionarily conserved mechanism [68,69].

Analysis of mRNA expression data [91,92,93,211] of validated or suspected cullin-4 substrate adapters suggests important roles for several known and uncharacterized Dcafs in cardiac and skeletal muscle development and function (Figure 6B,C). Examples of adapters with unique expression patterns include the small G-proteins Gnb1 and Gnb2, Dcaf6, Dcaf8, and Arpc1a (actin related protein 2/3 complex subunit 1a).

### 6.1. Cereblon

Cereblon (Crbn) is expressed mainly in the brain but is also found in cross-striated muscle tissues [260]. Crbn interacts with cullin-4A/B-Ddb1 in order to form a functional E3-ligase and regulates the levels of chloride voltage-gated channel 1 (Clcn1, also known as Clc1) [261]. Clcn1 channels are the mediators of skeletal muscle membrane excitability through depolarization of these cells [261]. Mutations in the *Clcn1* gene lead to myotonia in humans [262].

Crbn also directly regulates the catalytic alpha subunit of AMP-activated protein kinase (Ampk), a key protein for energy homeostasis [264], through the ubiquitin-dependent proteasomal degradation of Ampk-γ [265]. This action of Crbn is thought to be responsible for the attenuation of ischemia/reperfusion injury in *Crbn* knockout hearts [266].

Thalidomide, the notorious sedative to treat morning sickness that resulted in severe birth defects, and its analogs have been demonstrated to block Crbn functions [71,267,268] (Table 1). The drug has been recently repurposed as a promising anti-cancer drug by regulating the degradation of ikaros (also known as ikaros family zinc finger 1 (Ikzf1)) and aiolos (also known as ikaros family zinc finger 3 (Ikzf3)) transcription factors in multiple myeloma, which are required for growth of the tumor [269,270]. The positive effect of thalidomide on tumor repression is accompanied by the preservation of skeletal muscle fibers by reducing muscle expression of tumor-necrosis factor Tnf-α and transforming growth-factor Tgf-β, thereby preventing cancer-related cachexia [271].

Use of the drug as heart-failure treatment in humans and acute myocardial injury in rats suggested mild improvements of left ventricular ejection fraction and quality of life [272], as well as reduced myocardial necrosis and myofibril damage [273]. However, dose-limiting toxicity was observed in some patients, which exhibited bradycardia and acute renal failure among other adverse effects [272]. In addition, decreases in arterial pressure were observed in the rodent study, which may be attributed to depressant effects on the sino-atrial node [273].

### 6.2. Dcaf1

Dcaf1 is expressed in tissues that are important in the control of energy homeostasis [274]. The activity of this cullin-4B-Ddb1 adapter protein can be modulated by interaction with Sirt7, a member of the NADP-dependent deacetylases of the sirtuin family [275]. Studies of *Sirt7* knockout mice and knockdown experiments demonstrated that Sirt7 positively regulates the levels of the nuclear receptor Tr4/Tak1. Targets of Tr4 play essential roles in lipid homeostasis, fatty acid uptake and triglyceride synthesis/storage [276]. The authors of this manuscript show that Sirt7 regulates Tr4/Tak1 levels through the cullin-4B-Ddb1-Dcaf1 complex [275]. Later studies indicate that Sirt7 acts more generally on cullin-4-containing E3-ligase complexes beyond Dcaf1, as Sirt7 deacetylates the linker protein Ddb1 and inhibits its function [70]. However, to our knowledge pharmacological modulation of Sirt7 activity has not been tested as an approach to alter the cullin-4 E3-ligase function in muscles.

### 6.3. Dcaf6/Nrip

Dcaf6 (also called Nrip (nuclear receptor interaction protein)) is downregulated in patients with limb-girdle muscular dystrophy (LGMD). Mice lacking this calcium/calmodulin-binding protein display reduced myogenin expression, increased muscle weakness, and impaired exercise performance [277,278]. On the cellular level, Dcaf6 binds to muscle and non-muscle actinin proteins (Actn2, Actn1, and Actn4), and participates in calcium-dependent Nfatc1 (nuclear factor of activated T cells 1) signaling, and regulates mitochondrial activity and SR calcium homeostasis [278,279]. In addition, Dcaf6 plays in conjunction with myogenin a role in the postsynaptic development of the neuromuscular junction [277]. Besides its role in skeletal muscle, Dcaf6 also plays important functions in cardiac development. Loss of Dcaf6 results in progressive cardiac hypertrophy that ultimately leads to a dilated cardiomyopathy phenotype [279]. Hearts from knockout mice displayed impaired cellular calcium handling, a disruption of their sarcomeric structure, and aberrant mitochondrial architecture [279].

### 6.4. Dcaf8

Studies revealed that the expression of Ddb1-cullin-4-associated factor 8 (Dcaf8) is upregulated upon muscle denervation, following the expression pattern of the muscle ring finger protein MuRF1, a well-known atrogene that mediates muscle atrophy [280,281]. Cullin-4-Ddb1-Dcaf8 interacts with MuRF1 to modulate atrophy through the degradation of sarcomeric myosin. Indeed, C2C12 cells lacking Dcaf8 are resistant to glucocorticoid-induced myotube atrophy [280].

Mutations in *Dcaf8* cause axonal hereditary motor and sensory neuropathy (HMSN2) that results in atrophy of peroneal muscles and may be accompanied by cardiomyopathy [282].

### 6.5. Fbxo44

Unexpectedly, Fbxo44, an F-box protein that usually interacts with the cullin-1-Skp1 complex to form a functional E3-ligase, also associates with cullin-4B to mediate degradation of the regulator of G-protein signaling Rgs2 [160]. Rgs2 is highly expressed in the cardiovascular system and the central nervous system [283]. Rgs2 acts as an inhibitor of G-protein coupled receptor (GPCR) signaling pathways and is known to mediate vasoconstriction [284]. Expression of Rgs2 is found deregulated in cardiovascular diseases [285].

### 6.6. Gβ-Proteins Gnb1, Gnb2, and Gnb4

Several G-protein β-subunits were found to associate with Ddb1-cullin-4A E3-ligases independently of Gγ. This non-canonical function of Gβ-proteins was demonstrated to modulate G protein-coupled receptor kinase 2 (Grk2) protein stability and GPCR signaling via adrenergic receptors in cardiac muscles [68,254]. Intriguingly, loss of cullin-4A-dependent Grk2 regulation in *cullin-4A* knockout mice is linked to a sex-dimorphism: only male mice are affected by cardiac hypertrophy. However, the phenotype can be partially rescued in a heterozygous *Grk2* knockout background [68].

Clinically, a polymorphism in Gβ3 (C825T) is associated with hypertension, atherosclerosis, cardiac hypertrophy, stroke, and myocardial infarction. Patients also suffer from additional diseases, such as diabetes and insulin resistance. An intriguing study found that the polymorphism affected the binding of Gβ3 to the cullin-4 linker protein Ddb1, thereby again modulating GPCR signaling in affected patients [254].

### 6.7. Other Dcafs

As shown in Figure 6B,C, an array of additional cullin-4 substrate adapters display unique cardiac or skeletal muscle expression patterns. Among those, Arpc1A (actin-related protein 2/3 complex subunit 1A) is found upregulated in cardiac development but reduced in mRNA levels during skeletal muscle maturation. Aprc1A forms part of the Arp2/3 protein complex, which is important for actin polymerization and integrin signaling [286]. Intriguingly, levels of this Arp2/3 complex subunit were increased in cardiomyopathic *Fhl1* (four and a half lim domains 1) knockout hearts and hearts of patients with *Fhl1* mutations [287].

Besides the Fbxw7-based cullin-1 E3-ligase complex, levels of the mammalian target of rapamycin (mTOR) are also regulated by a cullin-4B-Ddb1 complex that utilizes raptor (Rptor) as substrate adapter protein [288]. The mTOR pathway, which involves the two different signaling complexes mTORC1 and mTORC2, is a master regulator of cell metabolism, growth, proliferation, and survival in cells, including cross-striated muscles [289]. This modulation of mTOR is antagonized by the ubiquitin hydrolase Uchl1, whose function balances signaling via mTORC1 and mTORC2 [290]. Loss of Uchl1 in patients was associated with the development of hypertrophic cardiomyopathy.

The mTOR pathway is also regulated by the Dcaf protein and autophagy master regulator Ambra1 (autophagy and beclin 1 regulator 1; also known as Dcaf3) [291]. Levels of Ambra1 are dynamically regulated through cullin-4, and dissociation of the complex through autophagic stimuli leads to the stabilization of this Dcaf protein. The stabilized Ambra1 then serves to indirectly limit mTOR signaling by inhibiting the cullin-5 dependent degradation of deptor (Dep domain-containing mTOR interacting protein) [292]. Silencing of Ambra1 in zebrafish negatively affected cardiac and skeletal muscle development, with smaller hearts and pericardial edema [293], as well as impaired muscle development, misaligned myofibers, and reduced locomotor activity [294]. Ultrastructural analysis of morpholino mutant fish showed sarcomeric disorganization, altered thin-filament lattice, scattered mitochondria, and defects of the sarcoplasmic reticulum. Mammalian Ambra1 is highly expressed in the nervous system, skeletal muscles, and heart [295], and knockout mice exhibit an early embryonic lethality with neural tube defects.

## 7. Functions of Cullin-5 and Its Substrate Adapters in Heart and Muscle

Similar to cullin-2, cullin-5 associates with the elongin-B/C heterodimer to attract substrate adapter proteins (Figure 1A and Figure 4A). Cullin-5-based adapters contain the B/C-box and Cul5-box motifs to form an ECS-type (elongin-B/C-cullin-Socs box) E3-ligase complex [178] (Figure 4A bottom panel). While the B/C-box motifs are similar in amino acid sequence between cullin-2 and cullin-5 adapter proteins, the adjacent Cul5-box is characterized by the relatively conserved LPΦP-motif, which is indispensable for binding to the cullin-5 scaffold [176].

Loss-of-function models suggest that cullin-2 and cullin-5 perform similar, yet distinct functions. Studies in *Drosophila* found that lack of cullin-5 results in NMJ defects in a subset of motorneuron terminals, where this E3-ligase is enriched. Double mutants that lack both cullin-2 and cullin-5 display a more severe phenotype compared to single mutants [183]. In mammals, cullin-5 functions for neuronal development are closely linked to the RING-finger protein Rbx2 (also known as Rnf7 or Roc2), rather than Rbx1 that is utilized by the majority of cullin E3-ligases [176,296].

*Cullin-5* knockout mice die in utero early in pregnancy, a result that is similar to other members of the cullin family. Heterozygous mice displayed no growth abnormalities [297]. However, to date, no muscle-specific knockouts of *cullin-5* have been generated.

Several substrate adapters for cullin-5 have been identified, mostly based on sequence homology of the conserved Socs-box motif [178], some of which display characteristic muscle-specific expression patterns (Figure 4B,C; Appendix A). This list, however, includes a mix of bona fide and suspected adapters and is not conclusive, as many novel adapters with less conserved Socs-box motifs may not be included.

### 7.1. Adapters of the Suppressor of Cytokine Signaling (Socs) Family

Proteins of the suppressor of cytokine signaling (Socs) family count among the best-characterized cullin-5 substrate adapters. Main functions for this protein family are the regulation of cytokines, such as interferon-γ (Ifn-γ), which are involved in the body’s immune response. Surprisingly though, not all Socs-protein family members form high-affinity E3-ligase complexes with cullin-5. Socs1 was shown to preferentially act as a substrate adapter for cullin-2 due to an incompletely conserved Cul5-box motif [176,298].

Despite the fact that the interactions between cullin-5, elongin-B/C, and Socs1 and Socs3 are the weakest compared to the other members of the family [299], it is interesting to note that this ECS complex has been identified as a therapeutic target to fight cachexia due to its role in Irs-1 degradation [300].

Socs-dependent signaling and potential degradation of substrate proteins also play a role in cardiac development and disease. *Socs3* knockout mice develop severe cardiac hypertrophy associated with myofiber disarray, resulting in perinatal lethality of mice. Intriguingly, additional loss of leukemia inhibiting factor receptor (Lifr) rescued the lethality phenotype [301]. In addition, Socs1/3 signaling plays a major role in the innate immune response of cardiomyocytes to enteroviral infections. Transgenic Socs1 mice with cardiac-specific overexpression of this Socs-family member successfully inhibited downstream signaling via janus kinase (Jak) and signal transducers and activators of transcription (Stat), thereby preventing viral replication and the accompanying myocarditis [302].

Similarly, Socs2 was found to play a vital role in immune-response and cardiac health in mice infected with *Trypanosoma cruzi*, a model for chagasic heart disease. Loss of Socs2 resulted in impaired cardiac function in infected mice, accompanied by altered calcium handling and reduced outward potassium channel function [303]. Socs2 signaling also plays a role in ischemia/reperfusion injury in type-2 diabetes mellitus. Levels of this Socs family member were markedly elevated in diabetic hearts and increased further after ischemia/reperfusion. Again, the suggested mechanism of action was via inhibition of Igf1 through the Jak/Stat signaling pathway [304].

Loss of Socs2 also had a profound effect on skeletal muscles, which is associated with the high-growth (Hg) phenotype in mice. Hg-mice and *Socs2* knockouts display delayed myogenesis, muscle fiber hyperplasia, and a subsequent increase in muscle mass. As a molecular mechanism, the deregulation of growth hormone and Igf1 secretion was shown to play a role in the development of this phenotype [305].

Several protein substrates have been identified, whose levels depend on Socs-adapter mediated degradation by the cullin-2/5-elongin-B/C E3-ligase complex. Examples include janus kinases 1 and 2 (Jak1, Jak2), insulin receptor substrate 1 and 2 (Irs1, Irs2) as well as the insulin receptor (InsR), and protein tyrosine kinase 2β (Ptk2B, also known as Pyk2) [306]. However, no muscle-specific substrates are known.

### 7.2. Cullin-5 Adapters of the Ankyrin Repeat and Socs-Box Containing (Asb) Protein Family

Cullin-5 substrate adapter proteins of the Asb-protein family contain besides the Socs-box multiple ankyrin repeat domains, which mediate interactions with a variety of putative substrate proteins. Several Asb-family members display characteristic expression in skeletal and cardiac muscles, including Asb1, Asb2, and Asb11 (Figure 4B,C). Asb2 (ankyrin repeat and Socs-box-containing 2), a component of ECS-type E3-ligase complex, has been shown to be essential for cardiomyocyte maturation in a zebrafish model [307]. Mutant fish fail to terminally differentiate, resulting in contractility and cardiac output defects. Substrates for Asb2 include Tcf3 (transcription factor 3), a bHLH transcription factor with roles in stem-cell proliferation and cardiovascular differentiation [308]. Crucial functions for this cullin-5 substrate adapter were also validated for mammalian cardiogenesis. Loss of Asb2 in mice resulted in embryonic lethality, characterized by reduced angiogenesis and underdeveloped hearts that displayed disorganized myofibrils and contractility problems [309]. The authors could further discern that the phenotype is due to the loss of the embryonic Asb2-α isoform of the protein in hearts, rather than the adult Asb2-β isoform. Splicing of Asb2 is also modulated by the RNA modifying enzyme Prmt1. Similar to the cullin-1 substrate adapter Fbxo40, hearts of mice that lack Prmt1 suffer from dilated cardiomyopathy and show aberrant Asb2 splicing [108]. Analysis of substrates for this cullin-5 adapter revealed its pivotal function to modulate filamin-A (Flna) protein levels, subsequently negatively affecting actin remodeling during cardiomyocyte development [309]. Asb2 was also demonstrated to ubiquitylate talin-1 (Tln1) [310], a protein that is together with talin-2 required for myoblast fusion and sarcomere assembly of skeletal muscles [311], and whose loss results in costameric instability and dilated cardiomyopathy in hearts [312].

Intriguingly, Asb2 has been shown to form a non-canonical E3-ligase with cullin-1 and cullin-4A, partially to regulate Jak2 protein degradation, a function that is more akin to Socs-family adapters of cullin-5 E3-ligases [313,314].

Other proteins of the Asb-family with high expression in skeletal and cardiac muscles are Asb3 and Asb8 [315,316]. Asb3 specifically mediated ubiquitylation of tumor necrosis factor receptor 2 (Tnfr2; also known as Tnfrsf1b (TNF receptor superfamily member 1B)) [315]. Tnf receptors bind tumor necrosis factor alpha (Tnf-α), which has been demonstrated to have cardio-depressant properties in heart-failure or ischemic injury [317,318,319].

Other potential substrates for Asb3 were Fas activated serine/threonine kinase (Fastk) or survivin (also known as Birc5 (baculoviral IAP repeat-containing 5)) [315]. Fastk was found upregulated in the hibernating myocardium of patients with ischemic left ventricular dysfunction [320]. Expression of survivin during embryonic development closely regulates overall cardiomyocyte number [321], and its expression in adult heart serves as a biomarker for cardiovascular risk and predictor of heart failure [322,323].

Asb11 is suspected to be important for the proliferation and maintenance of muscle progenitor cells in zebrafish [324]. The protein is evolutionarily related to the human Asb11 and Asb9 proteins, whose suspected protein substrates include brain and mitochondrial-type creatine kinase, or Talin-1 [306,313].

### 7.3. Elongin-A

The transcription and elongation factor elongin-A (also known as transcription elongation factor B polypeptide 3 (Tceb3)) displays a ubiquitous expression pattern, although skeletal muscle is among the tissues with the highest elongin-A mRNA levels [325]. Elongin-A binds to the elongin-B/C heterodimer to form the RNA polymerase II transcription elongation factor involved in mammalian cellular stress responses, and a cullin-5-based E3-ubiquitin ligase complex [326,327]. Loss of elongin-A in mice results in lethality between embryonic days E10.5 and E12.5 [328]. Knockout embryos and knockdown cells show premature senescence, presumably via the involvement of p38 Mapk and p53 signaling pathways. Embryos exhibited severe cardiac hypoplasia with abnormally thin myocardial walls that sometimes consisted of only one cell layer. Further understanding of the role of elongin-A for cardiac or skeletal muscle development is lacking. It is also unclear if the phenotype of elongin-A deficiency emanates more from its canonical functions as cullin-5 E3-ligase substrate adapter or non-canonical functions as RNA polymerase elongation factor.

### 7.4. Neuralized E3-Ubiquitin Protein Ligase 2 (Neurl2)

The Socs-box-containing adapter protein Neurl2 (also known as Ozz-E3) is specifically expressed in heart and skeletal muscles (Figure 4B,C) and is found upregulated during muscle development and regeneration, but not muscle atrophy [329,330]. Neurl2 forms a functioning E3-ligase complex with cullin-5, the elongin-B/C heterodimer, and Rbx1 with important roles during myofibrillogenesis. Knockout mice for this adapter protein develop sarcomeric abnormalities, including Z-disk streaming, increased frequency of myofibril-splitting, and centralization of nuclei in soleus and quadriceps muscles. Analysis of Neurl2 interaction partners revealed β-catenin as a putative substrate for this E3-ligase, a finding that was later confirmed by in vitro ubiquitylation assays and increased membrane-bound levels of the protein in knockout muscles. The same group of authors also revealed additional substrates for this cullin-5 adapter protein, including embryonic myosin heavy chain and Aip1 (also known as Pdcd6ip (programmed cell death 6 interacting protein)) [331]. Aip1 modulates the actin cytoskeleton in developing muscle fibers, and Neurl2 loss affects the formation of filopodia-like structures, myoblast migration, and adhesion [331]. Binding of Neurl2 to embryonic myosin is mediated by the C-terminus of the thick filament protein, localizing this complex at the sarcomeric M-band. Loss of Neurl2 prevents the downregulation of the embryonic myosin isoform during later stages of myofibrillogenesis, leading to its prolonged-expression.

## 8. Cullin-7/p193/p185 and Cullin-9/Parc Functions in Cross-Striated Muscles

Cullin-7 (also known as p193, p185) and cullin-9 (also known as p53-associated parkin-like cytoplasmic protein (Parc)) are two members of the cullin E3-ligase protein family with high sequence homology and unusual domain organizations. While all other cullin proteins contain N-terminal cullin-repeats, a centrally located cullin homology (CH)-domain that associates with the RING-domain proteins Rbx1 or Rbx2, and a C-terminal region that binds to E2-enzymes (Figure 1B), cullin-7 and cullin-9 contain several other domains (Figure 7A) [7]. Both proteins exhibit relatively high sequence homology and have instead of the cullin-repeats a CPH domain (conserved in Cul7, Parc, and Herc2) and an anaphase-promoting complex-like DOC1-domain (also called Apc10-domain, anaphase-promoting complex subunit 10). Both domains are also present in the Hect and Rld-domain-containing E3-ubiquitin protein ligase 2 (Herc2). Cullin-9 contains in addition to the CPH and DOC1 domains two C-terminally located RING-domains and an IBR-domain (in-between-RING-domain) [7]. This domain architecture is also known as the triad-supradomain. The appearance of this supradomain suggests that cullin-9 may not require the assistance of the RING-domain-containing Rbx1 or Rbx2 proteins to function as E3-ligase. However, high-throughput proteome data suggest that Rbx1 is still found in cullin-9-containing protein complexes [332]. Besides cullin-9, another E3-ligase that contains the triad supradomain is parkin, which is known to be important for mitophagy [333,334].

Loss of cullin-7 results in severe growth retardation during embryonic development. Knockout embryos were roughly 40% smaller than controls at embryonic day E18.5, and data seem to show that placental insufficiency may be the principal cause for the observed growth defect [335]. Lack of cullin-7 ultimately results in neonatal lethality due to breathing distress [335]. Functionally, cullin-7 was demonstrated to mediate proteasomal and lysosomal degradation of the mammalian Eag1 channel (ether a go-go 1, also known as Kcnh1 (potassium voltage-gated channel subfamily H member 1) and Kv10.1) [336], a voltage-gated potassium channel that was initially identified in *Drosophila eag* (ether a go-go) mutants, which display characteristic alterations to neuromuscular junction excitability [337]. The breathing distress observed in *cullin-7* knockouts is reminiscent of the neonatal lethality observed in skeletal muscle-specific *cullin-3* knockout mice [206], suggesting a possible muscle phenotype that may also extent to pathological changes in neuromuscular junctions. Despite these intriguing similarities, no thorough analysis of the muscle tissues in these embryos has been done to date, and defects in muscle formation/function (i.e., respiratory muscles) cannot be excluded.

Functionally, cullin-7-based E3-ligases are involved in cardiac tissue remodeling after cardiac infarction. The protein localizes exclusively to the cytoplasm of cells [338]. In vitro studies have shown that cullin-7 is involved in cardiomyocyte survival in E1A-induced cardiomyocyte apoptosis [339]. Although, cullin-7 is thought to present an alternative pro-apoptotic pathway to the p53 pathway in cardiac cells [339], both cullin-7 and cullin-9 are also known to interact with p53. Indeed, a study suggests that cullin-7 regulates cardiomyocyte cell cycle reentry in collaboration with the expression of the pro-apoptotic p53 protein following myocardial infarction (MI) [61]. These properties of cullin-7 were further exploited to probe beneficial effects in hearts to counteract adverse cardiac remodeling. Transgenic mice overexpressing a dominant-negative form of cullin-7 in hearts displayed reduced scar formation and better preserved cardiac function following MI [62]. Inhibition of cullin-7 activity in cardiomyocytes (Table 1) promotes proliferation at the infarct border zone and ventricular septum but has no effect on healthy cardiac tissue [60,61,62].

Based on these data, pharmacological inhibition of cullin-7-based E3-ligases could represent a potential strategy to treat myocardial infarction.

Cullin-9 is the last member of the cullin family. Cullin-9 shares high sequence homology with cullin-7 [340] and localizes in the cytoplasm of cells [341]. Cullin-9 also interacts with cullin-7 [342] and promotes the ubiquitylation and degradation of its substrates, like the survivin protein [341]. Unexpectedly, depletion of cullin-7 decreases the levels of survivin through a cullin-9 mediated mechanism. Forced expression of survivin rescued the phenotype in cells lacking cullin-7, indicating that cullin-7 is an inhibitor of cullin-9 E3-ligase activity [341].

Further experiments are required to better understand the role of cullin-9 for muscle tissues. Available data suggest a cullin-9-survivin pathway maintaining microtubule and genome integrity, as well as normal tissue development [341].

Very few bona fide substrate adapters have been reported for cullin-7 and cullin-9-based E3-ligases. This fact is underscored by a recent manuscript investigating cardiac functions for cullin E3-ligases [15]. While the authors were able to show that cullin-7 regulates the hippo signaling-pathway in cardiomyocytes, no substrate adapter has been reported to mediate the ubiquitylation of the hippo kinase Mst1 (also known as serine/threonine kinase 4 (Stk4)).

Nevertheless, one bona fide substrate adapter for cullin-7 has been reported: F-box and WD repeat domain-containing 8 (Fbxw8), a protein that belongs to the F-box domain protein family that is more associated with cullin-1 E3-ligase functions. As both cullin proteins heterodimerize, it may be possible that cullin-9 relies on Fbxw8-bound cullin-7 for its association with protein substrates. In addition, linker proteins have been reported that modulate cullin-7 functionality. Chief among them is obscurin-like 1 (Obsl1), a ubiquitously expressed protein with high sequence homology to the giant muscle-specific obscurin (Obsc) protein [343,344].

### 8.1. Fbxw8

Fbxw8 is perhaps the best-characterized cullin-7 substrate adapter. The protein forms a complex with Skp1, Rbx1, and cullin-7 to form an E3-ligase SCF complex [345]. A known function for the Fbxw8-containing cullin-7 E3-ligase complex is the regulation of Irs1 (insulin receptor substrate 1) [346,347]. The function of this signaling pathway protein is to transmit signals from the insulin-like growth factor 1 receptor (Igfr1) to downstream kinase pathways, such as Akt or Map-kinases. Skeletal muscle of *Igf1r* knockout mice present with reduced muscle fiber numbers and sizes [348]. Indeed, *Igf1*, *Igf2*, and *Igf1r* knockout mice present a very similar phenotype to *cullin-7* knockouts. Knockouts have a 55% reduction in their body mass and a dramatic decrease in their muscle mass [349]. Muscle-specific deletion of both insulin receptor substrates 1 and 2 (*Irs1/Irs2* double knockouts) severely decreased skeletal muscle growth, developed dilated cardiomyopathy, and caused death at 3 weeks of age [350]. Intriguingly, mTOR is required for degradation of Irs1 by modulating phosphorylation of the protein via p70 S6 kinase (also known as Rps6kb1 (ribosomal protein S6 kinase B1)). Ambra1, a regulator of mTOR function via cullin-4 and cullin-5 E3-ligases, also interacts with cullin-7. However, this function of the protein has not been further explored [292].

Besides Irs1, several other substrates have been identified, including the Golgi reassembly stacking protein 1 (Gorasp1, also known as Grasp65), and the plasma membrane protein Phldb2 (pleckstrin homology-like domain family B member 2; also known as LL5β) [332,351]. Gorasp1 plays an important function for Golgi morphology [351]. Fbxw8 regulates Gorasp1 levels, as loss of the F-box substrate adapter led to an accumulation of Gorasp1 [351]. Close spatial binding sites between Gorasp1, Gm130 (also known as golgin A2)—a structural protein localized to the cis-compartment of the Golgi-stack, and Herg potassium channels suggest a role of this Fbxw8 substrate for cardiac functions [352].

Phldb2 is a plasma membrane protein that plays a role together with cytoplasmic linker-associated protein 2 (Clasp2) in microtubule organization during neuromuscular junction development [353,354].

### 8.2. Btbd17b

Btbd17b (also known as Mac2 or Lgals3bp (lectin and galactoside-binding soluble 3 binding protein)) is a secreted protein that is elevated during the body’s immune response and interacts with components of the extracellular matrix system to facilitate tissue fibrosis [355]. This BTB-domain-containing protein has been found to associate with E3-ligases cullin-7, cullin-9, and Herc2, as well as Obsl1/Ccdc8-containing cullin-7 complexes [332,356,357]. While little is known about its functionality for cross-striated muscle development and function, Btbd17b is found deregulated in muscles of Duchenne muscular dystrophy patients [358] and involved in the immune-response of dengue-virus-infected satellite cells [359]. High levels of this protein were also found in the serum of patients suffering from Keshan cardiomyopathy [360]. However, whether Btbd17b is a bona-fide adapter protein for cullin-7 and/or cullin-9 remains to be determined.

### 8.3. Obsl1, Ccdc8, and the 3M-Growth Syndrome

Mutations in *cullin-7* are specifically associated with patients presenting the 3M-syndrome [361,362,363], an autosomal recessive disorder characterized by pre- and post-natal growth deficiencies that may involve muscles. People with 3M-syndrome present with an unusually short stature, characteristic facial features, and skeletal abnormalities. While *cullin-7* mutations are responsible for approximately 77.5% of 3M-syndrome cases [364], mutations in *obscurin-like 1* (*Obsl1*) and *Ccdc8* have also been shown to be associated with the syndrome [361]. Indeed, *Obsl1* mutations are estimated to account for 16.3% of cases [364]. Obsl1, a ubiquitously expressed protein with high homology to the N-terminus of the muscle-specific obscurin protein, interacts with the C-terminal region of cullin-7 [351] as well as cullin-9 [332]. Intriguingly, the interaction occurs in close proximity to the neddylation site in cullin-7 (Figure 7A). It is unknown if this posttranslational modification of cullin-7 affects Obsl1 binding. However, the association of Obsl1 and cullin-7 was shown to direct the E3-ligase to the Golgi apparatus, thereby acting as part of the subcellular anchoring mechanism. Loss of Obsl1 in neuronal cells resulted in altered Golgi morphology, but did not affect cullin-7 expression or localization [351]. Our data using murine knockout lung fibroblasts, and data from Obsl1 RNAi in Hek293 [365] indicate that Obsl1 is required for cullin-7 stability (Figure 7B). Surprisingly, cullin-1 protein levels were also reduced (Figure 7B), albeit to a lesser extent compared to cullin-7. Looking at mRNA levels of cullin-1, cullin-7, and cullin-9, as well as of the RING-domain protein Rbx1 we find compensatory upregulation upon loss of Obsl1 (Figure 7C). Combined, these data suggest that Obsl1 may have a broader impact on cullin E3-ligase regulation and activity than previously anticipated, although the exact mechanism remains enigmatic. Similar to Obsl1-RNAi of neuronal cells, the Golgi apparatus in murine lung fibroblast *Obsl1* knockout cells appeared more diffuse, as judged by Gm130, Csn5, and cullin-7 stain (Figure 7D,E). Loss of Obsl1 also resulted in reduced proliferation of cells, as shown by altered histone H3 phosphorylation (Figure 7B), possibly linked to altered cullin activity in *Obsl1* knockout cells.

Data indicate that Obsl1 is required as a linker for the association of coiled-coil domain 8 (Ccdc8) to cullin-7 [332,366]. Interaction of Obsl1 with Ccdc8 is additionally promoted by phosphorylation in the WW-domain of the coiled-coil domain-containing protein, mediated by casein kinase 2 (Ck2) and glycogen synthase kinase 3 (Gsk3). Obsl1 was also shown to promote the formation of a quaternary complex that additionally includes the ankyrin-domain-containing protein Ankra2 (ankyrin repeat family A member 2) [367]. Despite expression of Ankra2 and Ccdc8 in cross-striated muscles [366,368], there are currently no known muscle-specific functions for this quaternary cullin-7 E3-ligase complex. However, Ankra2 is known to interact with histone deacetylases Hdac4 and Hdac5, which have well-described functions in the heart and skeletal muscles [369]. Indeed, it was suggested that the quaternary complex mediates the degradation of these Hdacs [367].

Obscurin, the muscle-specific homolog of Obsl1, has also direct and indirect associations with cullin E3-ligases. Preliminary work from the lab of Guy Benian in *Caenorhabditis elegans* demonstrated that cullin-1 binds directly to unc-89, the nematode analog of obscurin (G. Benian, personal communication). The same group of investigators also reported that unc-89 participates in regulating Mei-1 (katanin) protein levels through binding to Mel-26, a cullin-3 and eIF3-associated BTB-domain-containing protein [370,371]. We characterized a similar mechanism through which binding of the mammalian obscurin protein to small ankyrin 1.5 (sAnk1.5) prevents its degradation through the Kctd6-containing cullin-3 E3-ligase complex [207].

In cross-striated muscles, both obscurin and Obsl1 are co-expressed and were shown to play important roles in maintaining muscle structure by binding redundantly to the sarcomeric proteins titin and myomesin [343,344]. Removal of both obscurin and Obsl1 affected sarcolemmal integrity, sarcoplasmic reticulum organization (including sAnk1.5 protein levels), and metabolism of skeletal muscles [372,373,374,375]. However, the reduction of cullin E3-ligase levels in tibialis anterior muscles is less noticeable compared to knockout lung fibroblasts (Figure 7F). This is specifically true for cullin-7, whose levels remain comparable to control muscles. The latter result may indicate tissue-specific differences of Obsl1 in modulating cullin levels, dissimilarities between deregulation of cullin protein levels in vitro and in vivo upon loss of Obsl1 (and/or obscurin), or the masking of cullin downregulation in myofibers due to other tissue types present in muscle that still express normal levels of the protein (e.g., fibroblasts or endothelial cells).

Despite the identification of mutations in *Obsl1* and its interaction with cullin-7 (and cullin-9), 3M-patients have not been described with gross muscle impairment or cardiomyopathy. Detailed characterization of 3M-patients’ cardiac and skeletal muscle function is still missing.

## 9. Cross-Talk of Cullin E3-Ligases

Numerous examples can be found in the literature that highlight the observation that none of the cellular protein degradation mechanisms works independently and autarkic from each other. In contrast, there is a great deal of interdependency and cooperativity between cellular degradation systems in general (i.e., the UPS and autophagy-lysosome system), and cullin E3-ligases in particular. Indeed, loss of functionality in one degradation system or E3-ligase leads typically to dynamic compensatory changes in the other degradation system, or changes in the activity of other E3-ligases, respectively. Examples for this dynamic interplay can be found in the inhibition of Csn5, which forms a crucial part of the Cop9 signalosome that critically modulates all cullin E3-ligase activity. Altered Csn5 levels led to compensatory changes in the expression of many cullin substrate adapter protein levels that counteract the now uninhibited cullin E3-ligase activity [54]. Another example is the regulation of the autophagy-lysosome system via cullin E3-ligases cullin-4 and cullin-5. Both cullins regulate the onset and termination of autophagy, respectively, by dynamically interacting with Ambra1, a regulator of autophagy [292].

Several cullin substrate adapters also display greater promiscuity towards their choice of cullin E3-ligase than anticipated. Prominent examples include Asb proteins or Fbxw8. The substrate adapter promiscuity towards cullin proteins may originate in the adapter protein domain layout, which can feature multiple domains that poise the adapter to be able to interact with several cullin family members; several cullin-4 substrate adapter proteins also contain Socs-box motifs (putative function as cullin-5 substrate adapter), F-box domains (putative function as cullin-1 substrate adapters), and RING-domains (with the possibility of independently acting as E3-ligases). Asb proteins form non-canonical E3-ligase complexes with cullin-1, cullin-2, and cullin-4, in addition to their well-known function as cullin-5 substrate adapter proteins [313,314]. Similar to Asb proteins, Fbxw8 associates with multiple cullin proteins [255,342,346,376], either via interaction through its F-box domain, or association through the cullin-4 specific linker protein Ddb1 [255]. Indeed, Fbxw8 is required for formation of a ternary cullin-1/cullin-7 complex to allow for proper placental development during embryonic development [376]. However, in the case of cullin-7, Fbxw8 was shown to prohibit formation of the cullin-7/cullin-9 heterodimer, presumably by occupying the same binding site [342].

Intriguingly, there are also substrate adapter- or linker protein-dependent feedback mechanisms that influence the levels of the associated cullin E3-ligase. Loss of Fbxw8 has, similarly to deletion of *Obsl1* (Figure 7B), effects on the expression levels of cullin-7 [345,377].

## 10. Future Directions of Research and Concluding Remarks

The breadth of muscle-specific roles for cullin-mediated protein turnover is just beginning to be known. However, results from knockout models, patients mutations, cardiac and skeletal muscle myopathies, and in vitro experiments underscore the crucial and at times surprisingly diverse functions of this E3-ligase family and its associated linker and substrate adapter proteins. Owing to the vast number of cullin-substrate adapter combinations and the largely unknown pool of muscle-specific substrates regulated by these CRL complexes, one can expect many more intriguing insights into their function for the biology of cross-striated muscles in health and disease. Specifically, the use of large-scale in vitro and in vivo proximity labeling approaches [134] in healthy and myopathic conditions will be of particular help to decipher the complex cullin E3-ligase degradome and the regulation of its activity.

### 10.1. Pathological Modulation of Cullin E3-Ligase Complexes May Underlie the Diversity of Myopathies and Clinical Phenotypes

The complicated spatiotemporal regulation of cullin substrate adapter expression and function may underlie one of the great mysteries for many myopathies: why certain skeletal muscle groups or fiber-types are more affected than others. Many substrate adapters (and their substrates) display extraordinarily diverse and complex muscle-group/fiber-type specific expression patterns (Figure 3D, Figure 4C, Figure 5F,G, and Figure 6C; and [211]). Hence, pathological mutations in a cullin-adapter protein or substrate may have more impact on muscle-groups with high expression levels of this adapter and/or its substrate, and spare muscles with lower protein levels of either. The same logic may also explain the finding that mutations in muscle genes that result in skeletal muscle myopathies have sometimes no or very limited impact on cardiac muscles (and vice versa). However, this hypothesis needs further investigation and may not fully explain all of the phenotypical peculiarities of cardiac and skeletal muscle myopathies.

### 10.2. The Emergence of Cullin E3-Ligases as Drug Targets and Clinical Perspectives

Pharmacological blockage of cullin activity has been of growing interest in the combat against cancer since it emerged that this class of E3-ligases serves as principal checkpoint on cyclin activity [8,9,141,142,145]. The use of Pevonedistat (Mln4924) as a therapeutic anti-cancer agent in clinical trials [24,25,26,27,28,29] to block all cullin activity underscores the importance of cullin E3-ligases as drug targets.

However, this non-selective approach was shown to come with considerable, albeit partially reversible, side-effects in cross-striated muscles [20,22,23,29]. Hence, considerable efforts are being made to search for more selective pharmacological molecules that only modulate the activity of a specific cullin isoform, substrate adapter, linker protein, or regulatory complex (Table 1). This approach should also lower unwanted side-effects. However, the search for more selective agents and the development and accreditation of this new class of drugs may take decades, given the enormous number and variety of cullin E3-ligase complexes and the intricate regulation of their activity. In addition, the emergence of extensive cross-talk within cullin E3-ligases, and with other protein degradation systems, complicates this quest for more selective molecules that specifically target activity of one CRL complex.

Either way, the use of highly selective CRL modulators would undoubtedly be of enormous benefit for patients suffering from cardiac and skeletal muscle myopathies, whose underlying pathology is caused by faulty cullin-dependent substrate (poly-) ubiquitylation.

## 11. Materials and Methods

### 11.1. Bioinformatics Analysis and Visualization

Hierarchical clustering using Pearson correlation was done using Morpheus (https://software.broadinstitute.org/morpheus/). Visualization of expression data was done using ImageJ, Excel (Microsoft, Redmond, WA, USA), and Photoshop (Adobe, San Jose, CA, USA). Visualization of structural PDB files was performed using WebLab Viewer 4.0 (Molecular Simulations Inc, San Diego, CA, USA) or Mol* [378] and RCSB PDB.

Generation of maximum likelihood molecular phylogenetic trees was done using MEGA7 [379,380]. In short, the evolutionary history was inferred by using the maximum likelihood method based on the JTT matrix-based model. The trees with the highest log likelihoods are shown. Initial trees for the heuristic search were obtained automatically by applying Neighbor-Join and BioNJ algorithms to a matrix of pairwise distances estimated using a JTT model, and then selecting the topology with superior log likelihood value. The tree is drawn to scale, with branch lengths measured in the number of substitutions per site. The analysis involved human F-box, BTB, or CH-domain sequences, available in Appendix A. All positions containing gaps and missing data were eliminated.

### 11.2. Cell Culture, Mln4924 Treatment, Immunofluorescence

Generation and culture of *Obsl1* knockout lung fibroblasts are described elsewhere [372]. Isolation and culture of neonatal mouse cardiomyocytes (NMCs) was done as previously described [381]. Treatment of cultured NMCs with 330nM Mln4924 (Santa Cruz Biotechnology, Dallas, TX, USA) or vehicle (dimethyl sulfoxide; Sigma-Aldrich, St. Louis, MO, USA) was done 48 h after plating for 48 h in maintenance medium.

Analysis of Mln4924 effects on C2C12 differentiation is described in detail elsewhere [20]. In short, C2C12 myoblasts at 80% confluency were transferred from growth medium (20% fetal calf serum, 1% penicillin/streptomycin, Dulbecco’s modified Eagle medium high glucose) to differentiation medium (2% horse serum, 1% penicillin/streptomycin, Dulbecco’s modified Eagle medium high glucose) containing either 330 nM Mln4924 or vehicle (dimethyl sulfoxide). Cells were differentiated for 5 days, with media replacements (differentiation medium with Mln4924 or vehicle) every other day.

Following treatment, cells were washed with 1x phosphate buffered saline (PBS) and fixed with 4% paraformaldehyde/PBS for 5 min at room temperature. To prepare immunofluorescence imaging, cells were permeabilized with 0.2% Triton X-100/PBS for 5 min and incubated with Gold buffer (20 mM Tris-HCl, pH 7.5, 155 mM NaCl, 2 mM ethylene glycol tetraacetic acid, 2 mM MgCl_2_, 1% bovine serum albumin) containing a mixture of primary antibodies in a humid chamber over night at 4 °C. Following overnight incubation, cells were washed three times with 1 × PBS at room temperature for 5 min each, and subsequently incubated with a mixture of fluorescently labeled secondary antibodies, fluorescently labeled-phalloidin (Life Technologies, Carlsbad, CA, USA) and/or DAPI (4′,6-diamidino-2-phenylindole; Sigma-Aldrich, St. Louis, MO, USA) dissolved in Gold buffer for 1 h at room temperature. Cells were washed three times with 1x PBS for 5 min, and mounted with fluorescent mounting medium (Agilent DAKO, Santa Clara, CA, USA). Fluorescently-labeled cells were imaged in sequential scanning mode on an Olympus Fluoview 1000 laser scanning microscope, equipped with a 40x or 63x oil immersion objective at zoom rates between 1 and 3. Image analysis was done using ImageJ and Photoshop.

### 11.3. Antibodies

The following primary antibodies were used for immunofluorescence and immunoblot analyses: Actn2 (clone EA53; Sigma-Aldrich, St. Louis, MO, USA), cullin-1 (C7117, Sigma-Aldrich, St. Louis, MO, USA), cullin-3 (C0871, Sigma-Aldrich, St. Louis, MO, USA; custom generated antibody, kind gift of Dr. Singer [8]), cullin-7 (sc-53810, Santa Cruz Biotechnology, Dallas, TX, USA; HPA030095, Sigma-Aldrich, St. Louis, MO, USA), Csn5 (sc-13157 and sc-9074; Santa Cruz Biotechnology, Dallas, TX, USA), Csn6 (BML-PW8295, Enzo Life Sciences, Farmingdale, NY, USA), Csn7a (sc-47968, Santa Cruz Biotechnology, Dallas, TX, USA), Csn7b (sc-47975, Santa Cruz Biotechnology, Dallas, TX, USA), Gm130 (610822, Clone 35/GM130, BD Transduction Laboratories, San Jose, CA, USA), and phospho-histone H3 (3377, Cell Signaling, Danvers, MA, USA). Secondary antibodies were from Jackson ImmunoResearch (West Grove, PA, USA) or Cell Signaling (Danvers, MA, USA).

### 11.4. Quantitative Real-Time PCR Analysis (qPCR)

For determination of mRNA levels, total RNA from cells was extracted using the Trizol reagent (Life Technologies, Carlsbad, CA, USA) according to the manufacturer’s instructions. First strand cDNA was generated from 2 µg of total RNA using random hexamers and Superscript II reverse transcriptase (Invitrogen, Carslbad, CA, USA). The following oligonucleotides were used to detect mRNA levels of murine cullin-1 (fwd: TTACAGCAGAACCCAGTTACTGAA, rev: ACTCTCCGCTGTTCCTCAAG), cullin-7 (fwd: CCTTCGGTCCCCCAATAC, rev: GCGGTGCAGGAAAAAGATT), cullin-9 (fwd: TTAGGACCCCATGACGACTC, rev: CAGGTACTCGTTCCAGCAAGA), Rbx1 (fwd: CGACAGACTGTGTGTTTCCAA, rev: GGCCACTGCATTCCACTTA). Reactions were done using PerfeCTa SYBR green real-time PCR mix (Quanta BioSciences, Beverly, MA, USA) on a CFX96 thermocycler (Bio-Rad Laboratories, Hercules, CA, USA). Samples were normalized to S18 levels (fwd: GGAAGGGCACCACCAGGAGT, rev: TGCAGCCCCGGACATCTAAG). All samples were run in triplicate (biological replicates).

### 11.5. Immunoblot Analyses

Preparation of samples and immunoblotting of proteins was done as previously described [382]. Samples were normalized to total actin. Uncropped original immunoblot images can be found in Appendix A.

### 11.6. Animals

All procedures involving vertebrate animals were reviewed and approved by the Animal Care and Use Committee at the University of California San Diego (protocol number S13009, approval date 12/19/2019). Skeletal muscle specific *obscurin* and *Obsl1* knockout mice were described previously [372].

### 11.7. Data Availability

Protein sequences used for generation of alignments and maximum likelihood molecular phylogenetic trees, and list of validated and putative cullin substrate adapters can be found in Appendix A. Original immunoblots can be found in Appendix A.

## Figures and Tables

**Figure 1 ijms-21-07936-f001:**
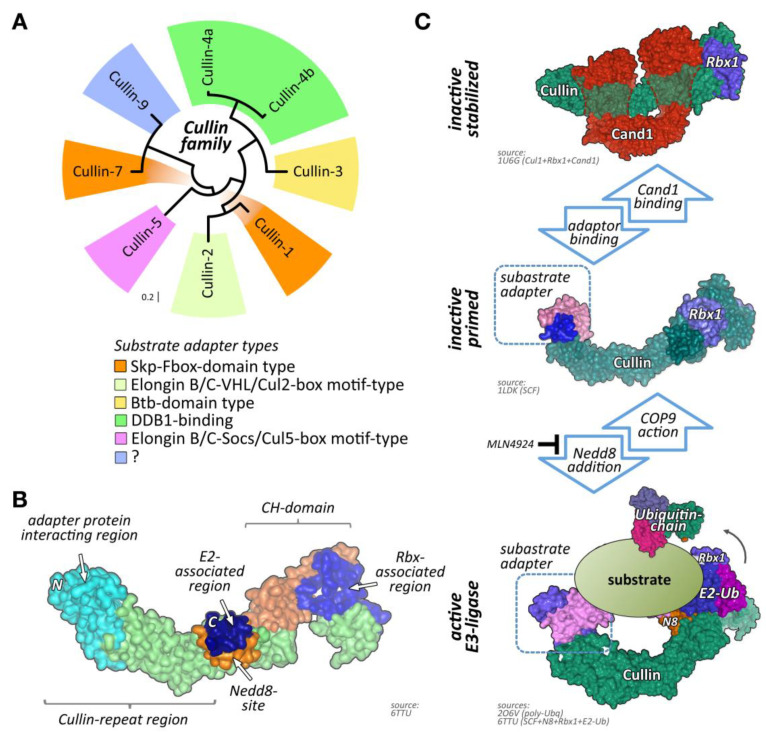
Cullin E3-ligases, domain overview, and activation. (**A**) Maximum likelihood molecular phylogenetic tree analysis of human cullin CH (cullin homology) domains. Shown are substrate adapter types for each of the cullin family members. (**B**) Schematics of cullin domain architecture, substrate adapter, and Rbx1 binding interfaces as well as the Nedd8-site in cullin-1. Surface representation of structure 6TTU [10]. (**C**) Simplified overview of cullin E3-ligase activation and substrate ubiquitylation. Inactive cullin bound to Rbx1 is stabilized by Cand1. Release of Cand1 allows for substrate adapter binding, “priming” the E3-ligase for a specific subset of substrate proteins. Adapter bound cullin gets fully activated by neddylation (N8), a step that is inhibited by Mln4924. Rbx1 associates with a ubiquitin loaded E2-enzyme. Binding of a substrate protein to the substrate adapter initiates the (poly-) ubiquitylation reaction. Cullin E3-ligases get inactivated by Cop9 signalosome action or enzymatic cleavage of Nedd8 by deneddylating enzymes Senp8 or Usp21. Surface representation of structures 1U6G, 1LDK, 6TTU, and 2O6V [10,11,12,13].

**Figure 2 ijms-21-07936-f002:**
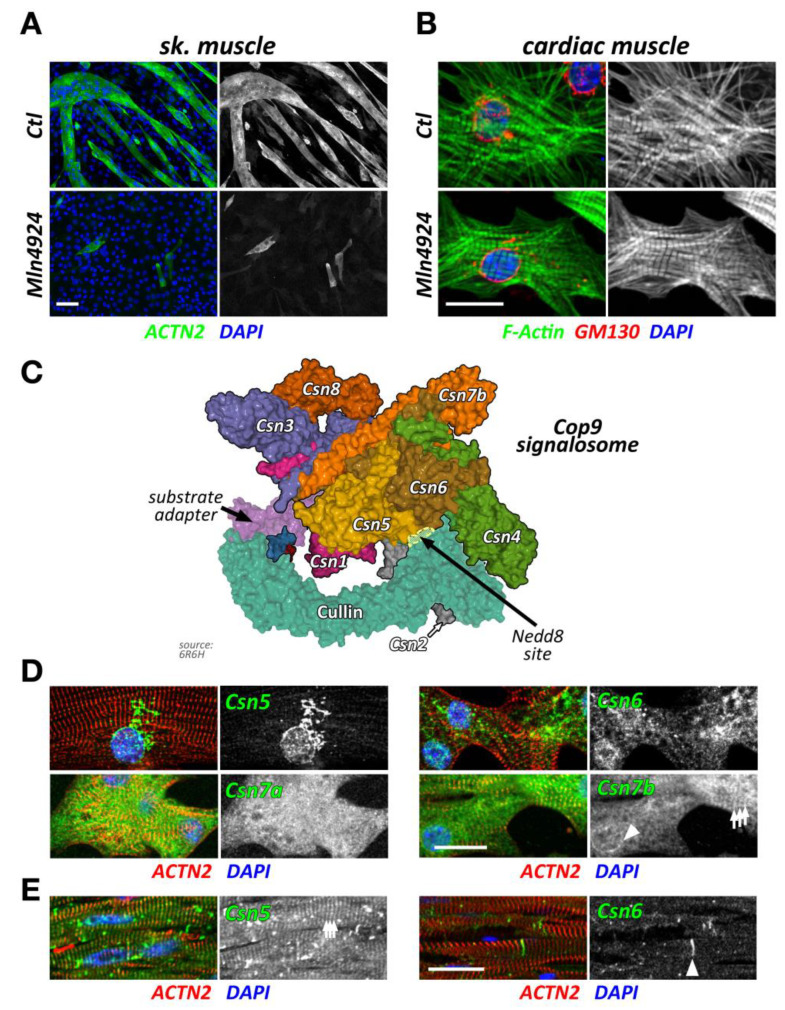
Cullin inhibition and characteristics of the Cop9 signalosome and its subunits. (**A**,**B**) Effect of cullin inhibition by Mln4924 on differentiating C2C12 cells (sk. muscle) after 5 days (also reported in [20]) (**A**) or neonatal mouse cardiomyocytes (cardiac muscle) after 2 days. C2C12 cells are decorated with antibodies against sarcomeric actinin (Actn2; green) and DAPI (blue). Neonatal mouse cardiomyocytes are stained with antibodies directed against GM130 (red) as well as phalloidin (green) and DAPI (blue). Scale bars = 20 µm. (**C**) Surface representation of the Cop9 signalosome complex (structure 6R6H [30]) together with cullin-1 bound to Skp1 and an F-box domain-containing substrate adapter. The site of neddylation in cullin is highlighted. The core subunits Csn5 and Csn6 are in close proximity to the neddylation site, while adjacent subunits Csn2 and Csn4 act as a clamp to correctly position the Rbx1 bound cullin protein for the deneddylation process. (**D**,**E**) Immunofluorescence analysis of Cop9 subunit localization in neonatal mouse cardiomyocytes (**D**) and adult mouse heart tissue (**E**). Core subunits Csn5 and Csn6 display localization to the endoplasmic reticulum or a diffuse localization in neonatal mouse cardiomyocytes (**D** upper panels), while they localize to the region of the Z-disk and intercalated disk in adult heart tissue (**E**; arrowhead denotes intercalated disk). Other peripheral Cop9 subunits, such as Csn7a or Csn7b also show divergent localization (**D** lower panels), displaying either diffuse (Csn7a), or perinuclear (arrowhead) and Z-disk-associated (arrows) targeting (Csn7b). Sarcomeric actinin (Actn2, in red) and DAPI (in blue) were used as counterstains. Scale bars = 20 µm.

**Figure 3 ijms-21-07936-f003:**
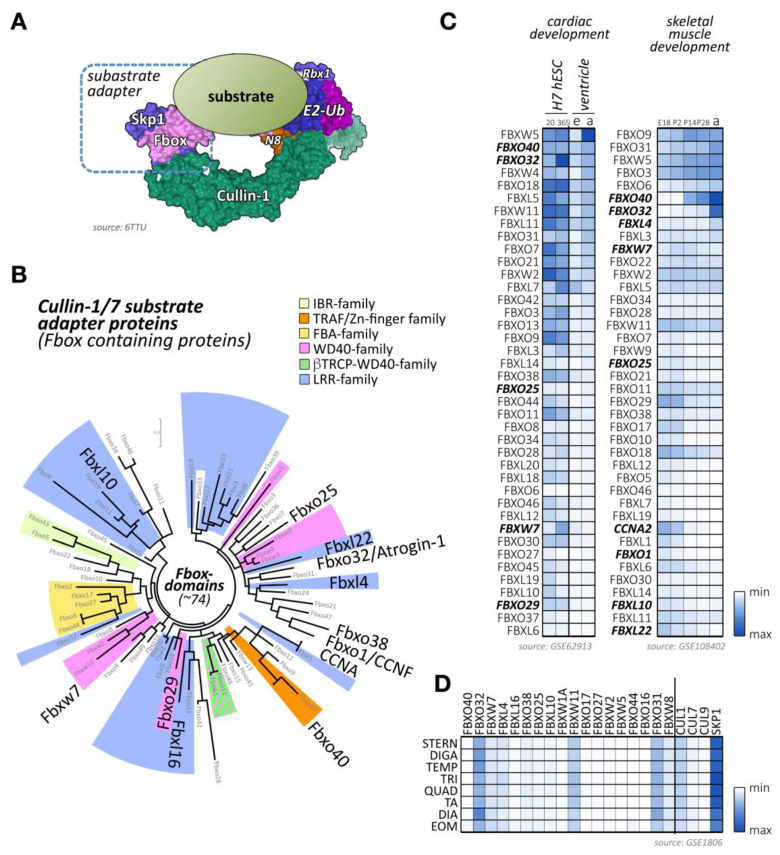
Cullin-1/7 substrate adapters of the F-box family. (**A**) Schematic view of a neddylated cullin-1 E3-ligase bound to Rbx1 and an Skp1-associated F-box protein, as well as a ubiquitin-loaded E2-enzyme. The substrate with a target lysine for (poly-) ubiquitylation is positioned by the fully assembled cullin-1 E3-ligase in close proximity to the enzymatically active Rbx1 and E2 proteins. Surface representation is from structure 6TTU [10]. (**B**) Maximum likelihood molecular phylogenetic tree analysis of human F-box domain-containing proteins. Adapter protein families for the cullin-1/7 E3-ligase, identified by their other domains are highlighted, as are F-box proteins with known functions for cross-striated muscles. (**C**) Heatmap analysis of F-box protein mRNA expression levels during cardiac development (differentiation of H7 human embryonic stem cells into cardiomyocytes at day 20 and day 365; embryonic (e) or adult (a) mouse ventricular tissue; left panel) and skeletal muscle development (at embryonic day E18, and postnatal days 2, 14, 28, and adult stages; right panel). Shown are the top 40 F-box-proteins with the highest mRNA levels in skeletal and cardiac muscles. Adapter proteins with known functions for heart or skeletal muscle are highlighted. Data were adapted from GSE62913 and GSE108402 [91,92]. (**D**) Muscle-type specific mRNA expression analysis of top 18 F-box-containing adapter proteins with known roles in cross-striated muscles, as well as cullin-1, cullin-7, cullin-9, and Skp1. Analyzed adult muscles were sternomastoid (Stern), digastric (Diga), temporalis (Temp), triceps (Tri), quadriceps femoris (vastus lateralis; Quad), tibialis anterior (TA), diaphragm (Dia), and extraocular muscle (Eom). Data were adapted from GSE1806 [93].

**Figure 4 ijms-21-07936-f004:**
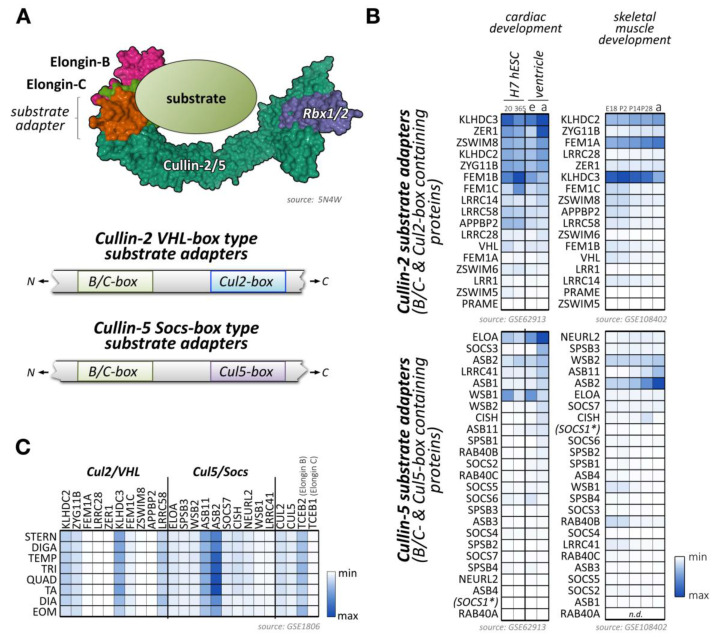
Known and potential substrate adapters and organization of cullin-2/5-based E3-ligases. (**A**) Organization of cullin-2/5 E3-ligases. Schematic overview of cullin-2/5 E3-ligase assembly. The cullin-2 or cullin-5 scaffold associates with RING-domain-containing Rbx proteins, and the elongin-B/C heterodimer that links to a VHL-box or Socs-box type substrate adapter, respectively (top panel). Schematic representation of VHL- and Socs-type substrate adapter interaction-motif organization (bottom panel). Surface representation is from structure 5N4W [181]. (**B**) Heatmap analysis of VHL-type (top panels) and Socs-type substrate adapter mRNA expression levels (bottom panels) during cardiac development (left panels) and skeletal muscle development (right panels). Left panels show analysis of cardiac differentiation of H7 human embryonic stem cells into cardiomyocytes, analyzed at day 20 and day 365 in culture, and ventricular tissue of mouse embryonic (e) or adult hearts (a). Right panels display the analysis of murine skeletal muscle differentiation at embryonic day E18, and postnatal days 2, 14, 28, and adult stages. Only the top 40 adapter proteins with the highest mRNA levels in cardiac and skeletal muscles are shown. Data were adapted from GSE62913 and GSE108402 [91,92]. The asterisk denotes that Socs1 has a comparatively low affinity for cullin-5, but a higher affinity for cullin-2, due to its unusual Cul5-box sequence. (**C**) Muscle-type specific mRNA expression analysis of the top 10 VHL- and Socs-box adapter proteins, as well as cullin-2, cullin-5, and the elongin-B/C proteins. Analyzed adult muscles were sternomastoid (Stern), digastric (Diga), temporalis (Temp), triceps (Tri), quadriceps femoris (vastus lateralis; Quad), tibialis anterior (TA), diaphragm (Dia), and extraocular muscle (Eom). Data were adapted from GSE1806 [93].

**Figure 5 ijms-21-07936-f005:**
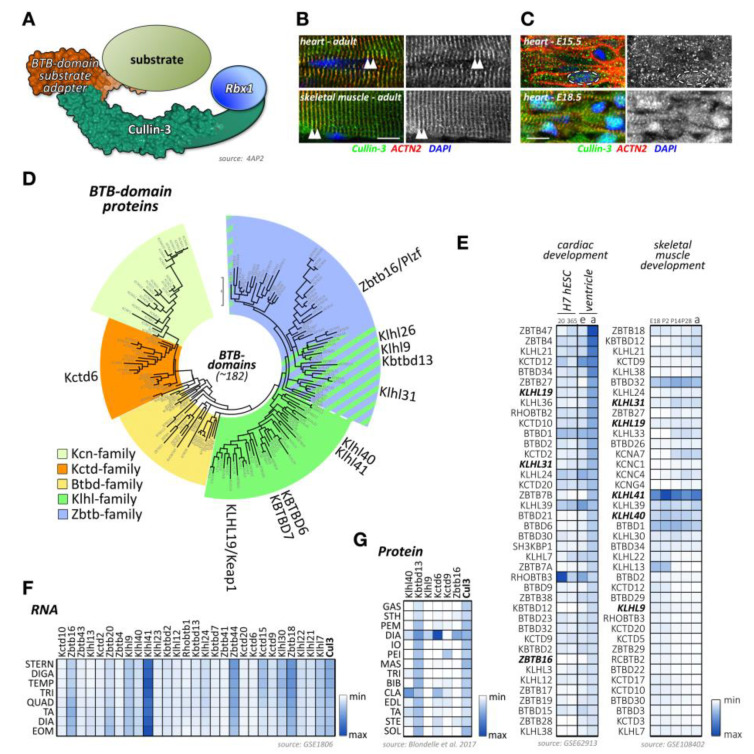
Cullin-3 and its substrate adapters. (**A**) Schematic presentation of the cullin-3-based E3-ligase with a BTB-domain-containing substrate adapter. Surface representation is from structure 4AP2 [210]. (**B**,**C**) Localization of cullin-3 (green) in mouse adult heart and skeletal muscle (**B**), and mouse embryonic heart at E15.5 and E18.5. (**C**) Tissues were counterstained with antibodies against sarcomeric actinin (ACTN2, red) and DAPI. Arrowheads highlight the sarcomeric Z-disk (B). Dashed line in (**C**) marks the boundaries of a single cardiac nuclei. Scale bar = 10µm. (**D**) Maximum likelihood molecular phylogenetic tree analysis of human BTB-domain-containing proteins. Shown are different BTB-domain-containing substrate adapter families. Cullin-3 substrate adapters with known functions for cross-striated muscles are highlighted. (**E**) Heatmap analysis of BTB-domain-containing substrate adapter mRNA expression levels during cardiac development (left panel) and skeletal muscle development (right panel). Left panel shows analysis of cardiac differentiation of H7 human embryonic stem cells into cardiomyocytes. Cells were analyzed at day 20 and day 365 in culture. Expression levels of adapter proteins in the ventricular tissue of mouse embryonic (e) or adult hearts (a) are also depicted. Right panels display analysis of murine skeletal muscle differentiation at embryonic day E18, and postnatal days 2, 14, 28, and adult stages. Note that only the top 40 proteins with the highest mRNA levels in cardiac and skeletal muscles are depicted. BTB-domain-containing adapter proteins with known functions for heart and/or skeletal muscles are highlighted in bold. Data were adapted from datasets GSE62913 and GSE108402 [91,92]. (**F**) Muscle-type specific mRNA expression analysis of BTB-domain-containing adapter proteins, as well as cullin-3. Analyzed adult muscles were sternomastoid (Stern), digastric (Diga), temporalis (Temp), triceps (Tri), quadriceps femoris (vastus lateralis; Quad), tibialis anterior (TA), diaphragm (Dia), and extraocular muscle (Eom). Data were adapted from GSE1806 [93]. (**G**) Protein expression heatmap of select cullin-3 substrate adapter proteins in various adult mouse skeletal muscles. The following muscle-types were analyzed: gastrocnemius (Gas), sternohyoideus (Sth), pectoralis major (Pem), diaphragm (Dia), internal oblique (Io), pectoralis minor (Pei), massester (Mas), triceps (Tri), biceps brachii (Bib), clavotrapezius (Cla), extensor digitorum longus (Edl), tibialis anterior (Ta), sternomastoideus (Ste), and soleus (Sol). Adapted from [206].

**Figure 6 ijms-21-07936-f006:**
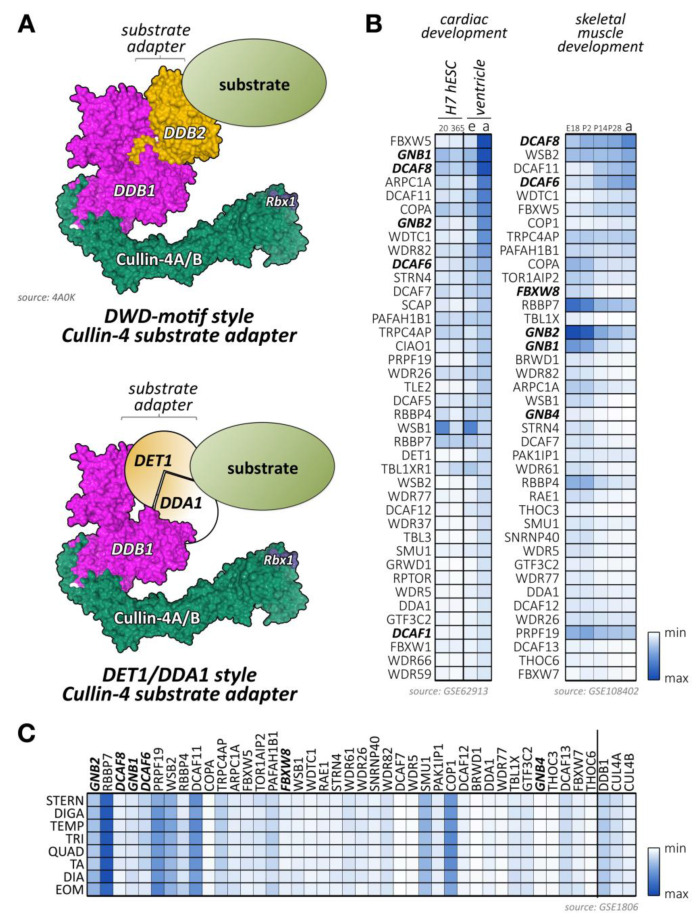
Substrate adapters for cullin-4A/4B-Ddb1 E3-ligases. (**A**) Complex organization of cullin-4A/B-based E3-ligases (surface representation from structure 4A0K [263]). Both cullin-4A/B scaffolds associate with the linker protein Ddb1 that binds to the various substrate adapter proteins. The most common adapters contain a DWD-motif mediating interaction with Ddb1 (top panel). Another adapter type is represented by the heterodimer of Det1 and Dda1 (bottom panel). (**B**) Heatmap analysis of known and suspected cullin-4A/B substrate adapter mRNA expression levels during cardiac development (differentiation of H7 human embryonic stem cells into cardiomyocytes at day 20 and day 365; as well as embryonic (e) or adult (a) mouse ventricular tissue; left panel) and skeletal muscle development (at embryonic day E18, and postnatal days 2, 14, 28 and adult stages; right panel). Data were adapted from GEO datasets GSE62913 and GSE108402 [91,92]. Adapter proteins with known functions for heart or skeletal muscles are highlighted in bold. (**C**) Muscle-type specific mRNA expression analysis of top 40 cullin-4A/B substrate adapters with known roles in cross-striated muscles, as well as the linker Ddb1, cullin-4A, cullin-4B itself. Analyzed adult muscles were sternomastoid (Stern), digastric (Diga), temporalis (Temp), triceps (Tri), quadriceps femoris (vastus lateralis; Quad), tibialis anterior (TA), diaphragm (Dia), and extraocular muscle (Eom). Data were adapted from GSE1806 [93].

**Figure 7 ijms-21-07936-f007:**
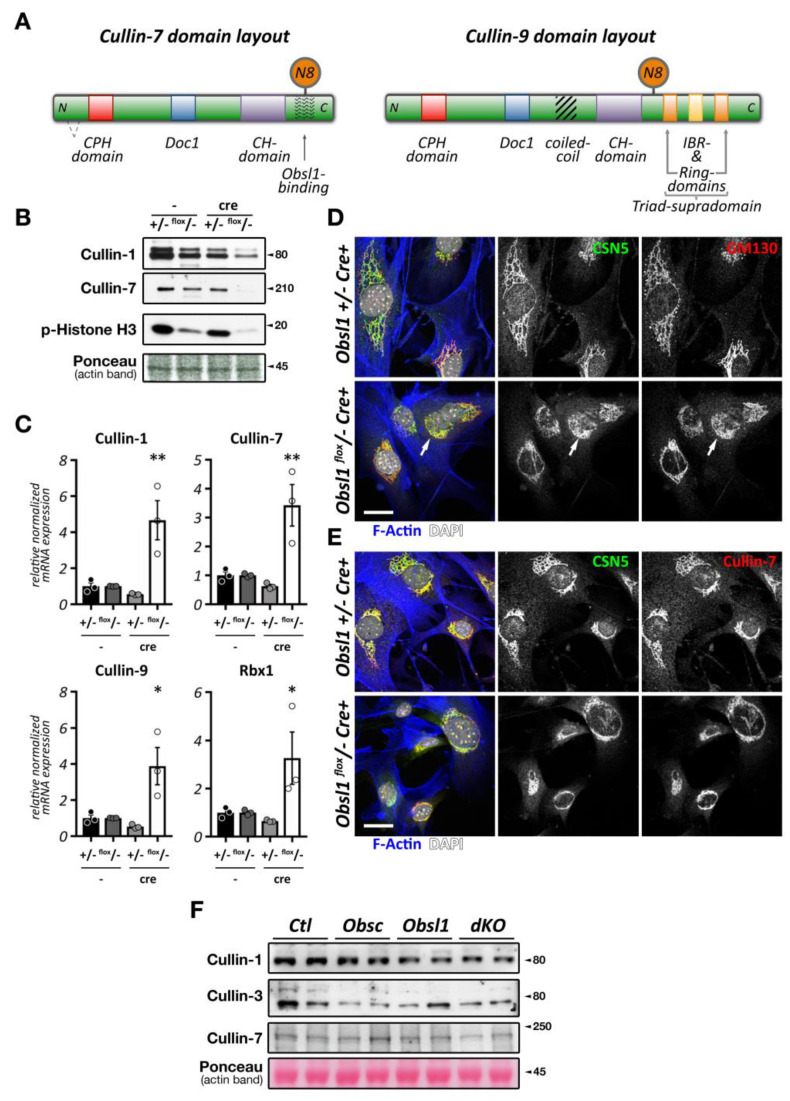
Cullin-7 and cullin-9 E3-ligases, and effects of Obsl1 for cullin E3-ligase expression and localization. (**A**) Cullin-7 and cullin-9 domain layout. Domain organization, mapped neddylation sites (N8), and Obsl1 interaction sites, as well as splicing in cullin-7, are shown. (**B**) Ablation of *Obsl1* in mouse lung endothelial cells results in a reduction of cullin-1 and loss of cullin-7 on the protein level. Cells also show reduction in levels of phosphorylated histone H3. Ponceau stained actin bands are shown as loading controls. (**C**) Analysis of cullin-1, cullin-7, cullin-9, and Rbx1 mRNA expression levels in mouse lung endothelial cells with genetic ablation of *Obsl1*. Expression was normalized to S18. * *p* < 0.05, ** *p* < 0.01 vs. +/- (cre -) controls as determined by ANOVA analysis. (**D**,**E**) Effect of *Obsl1* deletion on the localization of Cop9 subunit Csn5 (green in D,E) and cullin-7 (red in E). Cells were counterstained with GM130 (red in D), as well as fluorescently labeled phalloidin (blue in overlay) and DAPI (white in overlay). Arrow highlights areas of disorganized Golgi structure upon loss of Obsl1. Scale bar = 10µm. (**F**) Analysis of cullin-1, cullin-3, and cullin-7 protein levels in control (Ctl), *obscurin* knockout (Obsc), *Obsl1* knockout (Obsl1), or *obscurin/Obsl1* double knockout (dKO) tibialis anterior muscles. Ponceau stained actin bands are shown as loading controls. Unmodified blot images can be found in Appendix A.

**Table 1 ijms-21-07936-t001:** Pharmacological and experimental modulation of cullin-RING ligase (CRL) activity. This table summarizes some of the pharmacological and experimental methods that have been used to investigate and modulate cullin E3-ligase function. Detailed descriptions on the mechanism of action can be found in the main text and references.

Target	Compound/ Method	Effect	Reference(s)
Nae1	Mln4924 (Pevonedistat)	Nae1 inhibition and subsequent lack of cullin neddylation Deactivation of all CRL	[19]
Cop9 signalosome	Csn5i-3	Targeted inhibition of Csn5 deneddylation activity Keeps cullin in neddylated (active) state. Leads to inactivation of a subset of CRLs by inducing degradation of their adapter proteins	[54]
	Doxycycline	Inhibition of Csn5 activity.	[55]
Rbx1	Glomulin	Blocks Rbx1 from binding E2	[56]
Cullin	Co^2+^	Cullin-2 inhibition	[57]
	Cullin-2 elongin-B/C interaction inhibitor peptide	Inhibition of cullin-2 E3-ligase complex formation	[58]
	miR-424	Targets cullin-2 expression in endothelial cells	[59]
	Dominant- interfering cullin-7 mutants	Inhibits cullin-7 activity in transgenic mice with cardiac restricted expression of the mutant	[60,61,62]
Cullin substrate adapters and linker proteins	Oligonol	Indirect Fbxo32 downregulation via Sirt1 upregulation	[63]
	CI-994	Indirect Fbxo32 regulation via HDAC1 inhibition	[64]
	Roflumilast	Indirect Fbxo32 inhibition regulation through inhibition of phosphodiesterase-4 proteolysis	[65]
	Imperatorin	Indirect Fbxo32 downregulation by inhibition of Stat3 signaling pathway	[66]
	Ikkβ RNAi or abrogation of cIAP (cellular inhibitor of apoptosis 1) function LCL161 (smac mimetic compounds)	Indirect Fbxo32 downregulation by NFkB pathway inhibition or cIAP depletion	[67]
	H89 (PKA inhibition)	Modulation of PKA-dependent DDB1 phosphorylation at Ser645 and linked cullin-4 E3-ligase activity via Gβ proteins	[68,69]
	*Sirt7* knockout and knockdown	Sirt7 deacetylates Ddb1 and leads to inhibition of cullin-4-based E3-ligases	[70]
	Thalidomide and analogs	Inhibition of cereblon-based cullin-4 E3-ligases	[71]

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
