# Peer review of "The Role of Cullin-RING Ligases in Striated Muscle Development, Function, and Disease"

_ijms, 2020, doi:10.3390/ijms21217936_

Round 1

Reviewer 1 Report

The manuscript by Blondelle et al addresses in detail state of the art CRL. The content is very granular and well organized. Some minor suggestions are provided below.

1) This manuscript in very extensive. It will be easy for the readers to have a summary of contents with page numbers to look for specific topics. Also, it will help to organize the reading.

2) having the figures and figure legends in the same page will improve readability, particularly with the rich text found in legends.

3) Minor English editing is required.

4) Text in lines 277-281 is duplicated.

5) Table 1 should be fitted to be on one page.

6) This is for Figure 3-C, but applies to other figures too. It is hard to get the take home message from the heatmaps. It just a very extensive list of genes that may be more adequate to keep as a reference list in supplemental data.

7) It would be useful to add a future directions of research within the Concluding remarks section

Author Response

We thank the reviewers for their comments and suggestions.

1) This manuscript in very extensive. It will be easy for the readers to have a summary of contents with page numbers to look for specific topics. Also, it will help to organize the reading.

We agree tat the manuscript is quite extensive. The addition of a table of content is an excellent suggestion by the reviewer to help the reader find a topic of interest. Hence, we added a table of content at the beginning of the manuscript.

2) having the figures and figure legends in the same page will improve readability, particularly with the rich text found in legends.

We agree with the reviewer and formatted the manuscript accordingly.

3) Minor English editing is required.

We proof-read the manuscript and corrected any mistakes.

4) Text in lines 277-281 is duplicated.

We apologize for this mistake. This paragraph was intended as a table legend instead of appearing in the main text. We corrected this formatting mistake.

5) Table 1 should be fitted to be on one page.

We agree with the reviewer and formatted the manuscript accordingly.

6) This is for Figure 3-C, but applies to other figures too. It is hard to get the take home message from the heatmaps. It just a very extensive list of genes that may be more adequate to keep as a reference list in supplemental data.

We thank the reviewer for this comment. The heatmaps are designed for interested readers as a data-mining tool. Many of the genes/proteins listed in the heatmaps have no assigned functions in cardiac or skeletal muscles. The developmentally regulated expression of uncharacterized substrate adapters may be suggestive of crucial functions in the formation and/or maintenance of cross-striated muscles. As we specifically wanted to highlight these uncharacterized proteins, we included expression heatmaps for the most highly expressed adapter proteins for each Cullin protein family member. While extensive, moving the heatmaps to the supplemental data would make them less accessible in our view. To highlight our intention, we added a paragraph detailing future directions within the concluding remarks, as also suggested by the reviewer.

7) It would be useful to add a future directions of research within the Concluding remarks section

We thank the reviewer for this excellent suggestion and incorporated sections with future directions into the concluding remarks part of the manuscript.

Reviewer 2 Report

Manuscript ID: ijms-960652

Type: Review

Title: The role of Cullin-RING ligases in striated muscle development, function and disease.

Authors: Jordan Blondelle, Andrea Biju, Stephan Lange

This is an interesting and timely review by Blondelle et al. It is a very ambitious and comprehensive work on the role of cullin-RING ubiquitin ligases (CRLs) in striated muscle.

The introduction outlines the functioning of the ubiquitin proteasome system (UPS). Interestingly, in the striated muscle, about 80% of protein degradation is dependent on the UPS. The authors introduce cullin-RING ubiquitin ligases (CRLs), one of the largest group of ubiquitin E3 ligases. They describe common regulators of CRLs such as Nedd8 and neddylation, Cop9 signalosome (Csn), Nedp1 (deneddylating enzyme 1, Den1), the deubiquitylating enzymes Uchl3, Usp21 and the Cullin-associated neddylation-dissociated 1/2 (Cand1/2).

In the main part of the review, the authors summarize recent findings on the role of CRLs in protein degradation in striated muscle and add some of their own recent data. In chapter 3, CRL1 complexes possessing F-box proteins as substrate receptors (SR) are discussed. More than 10 F-box proteins, their specific substrates and role in cross-striated muscles are described. The authors present functions of CRL2s in cardiac and skeletal muscles. They discuss structural similarities between CRL2 and CRL5 complexes. The physiology and pathophysiology of the VHL-Hif1-alfa axis is discussed. Moreover, the authors show the surprising diversity and functions of CRL3 complexes and their adaptor proteins/SRs called BTB proteins. They analyze 7-8 BTB proteins and their specific substrates and functions in protein degradation in cardiac and skeletal muscles in detail. One of the most diverse family of CRLs are CRL4A/B complexes. There is a large number of Ddb1 and Cul4-associated factors (Dcafs) and in combination with other SRs a high diversity of CRL4A/B complexes with diverse substrate specificity has been identified in striated muscle. Cullin 5 and cullin 2 perform overlapping as well as distinct functions in muscle protein degradation. Socs proteins are main SR of CRL5 complexes and regulation of cytokines seems to be their main function in muscle cells as well. More recently, structure and function of CRL7 and CRL9 complexes has been studied, which is supported by own data of the authors. Interestingly, also CRL7 and CRL9 complexes seem to exert roles in the cross-striatal muscle.

The review is a very good overview on the role of CRLs in striated muscle development, function and disease. It underlines the significance of CRLs as pharmaceutical targets. It will be of interest to the broad readership of ijms and beyond.

Minor points

  • Although in general citations seem to be very detailed, the introduction part is rather weak. For example, the introduction of the UPS requires citation of the fundamental work by Hersko and Ciechanover. Moreover, the structural similarity between the Csn and 26S proteasome lid and the eIF3 complex was shown 10 years earlier by Glickman et al. (1998). A subcomplex of the proteasome regulatory particle required for ubiquitin-conjugate degradation and related to the COP9-signalosome and eIF3. Cell 94, 615-623 and by Seeger et al. (1998). A novel protein complex involved in signal transduction possessing similarities to 26S proteasome subunits. FASEB J. 12, 469-478.

Discussion on Csn activity and regulation of CRLs by Cand1/2 without citing the work of Deshaies and co-workers is incomplete.

  • Texts between lines 277-279 as well as 279-282 are identical.
  • Line 1045: “wit” should be “with”
  • Inconsistencies: Csn-CSN; Cop9-COP9; Mln4924-MLN4924; Hif1-alpha-HIF1-alpha; Cullin-4a-Cullin-4A.

Author Response

We thank the reviewers for their comments and suggestions.

Minor points

Although in general citations seem to be very detailed, the introduction part is rather weak. For example, the introduction of the UPS requires citation of the fundamental work by Hersko and Ciechanover. Moreover, the structural similarity between the Csn and 26S proteasome lid and the eIF3 complex was shown 10 years earlier by Glickman et al. (1998). A subcomplex of the proteasome regulatory particle required for ubiquitin-conjugate degradation and related to the COP9-signalosome and eIF3. Cell 94, 615-623 and by Seeger et al. (1998). A novel protein complex involved in signal transduction possessing similarities to 26S proteasome subunits. FASEB J. 12, 469-478.

We thank the reviewer for his/her insight and apologize for the brevity of the introduction. The focus of this review is the role of Cullin E3-ligases and their associated proteins for the development and function of cross-striated muscles. We agree that fundamental work on the UPS by a number of scientists in this extensive field has not been cited to concentrate on the topic of this review. Nevertheless, we now included several references (including works by Hershko and Ciechanover) that give the reader an opportunity to find more detailed descriptions of general UPS functions for further reading.

We also included the two works by Glickman and Seeger, highlighting similarities between the Cop9 signalosome and the proteasomal lid and iEF complexes. We thank the reviewer for alerting us to this omission.

Discussion on Csn activity and regulation of CRLs by Cand1/2 without citing the work of Deshaies and co-workers is incomplete.

We apologize for the oversight and included now references to the work of Deshaies and co-workers.

Texts between lines 277-279 as well as 279-282 are identical.

We apologize for this mistake. This paragraph was intended as a table legend instead of appearing in the main text. We corrected this formatting mistake.

Line 1045: “wit” should be “with”

We thank the reviewer for pointing out this mistake and corrected the spelling.

Inconsistencies: Csn-CSN; Cop9-COP9; Mln4924-MLN4924; Hif1-alpha-HIF1-alpha; Cullin-4a-Cullin-4A.

We corrected the inconsistencies in the manuscript, and thank the reviewer for pointing us to these specific instances.